# Preference learning made easy:
# Everything should be understood through win rate

**Lily H. Zhang**[1]   **Rajesh Ranganath**[1,2]

## Abstract

Preference learning, or the task of aligning generative models to preference comparison data, has yet to reach the conceptual maturity of classification, density estimation, etc. To close this gap, this work presents a framework to understand preference learning starting from the sampling distribution of pairwise preference data. First, we prove that the only evaluation of a generative model that respects both preferences and prevalences in the data distribution is a form of win rate, justifying win rate as the focal point to understand preference learning. We then analyze preference learning methods as win rate optimization (WRO) or non-WRO. We present novel instances of WRO beyond existing examples (RLHF, NLHF) and identify two key theoretical benefits of all such methods. We prove that common non-WRO methods like DPO and SFT on preferred samples lack these properties and suggest ways to mitigate such theoretical limitations. We also show that WRO underperforms in practice due optimization difficulties and that optimization success predicts performance better than choices which affect the objective's solution. Our analysis highlights best practices for existing methods and provides recommendations for future research, guided by the principle that one should either align non-WRO methods more closely with WRO or improve the optimization of WRO objectives.

## 1. Introduction

Learning from preference data, often referred to as human feedback, has emerged as a key step in training large language models, particularly given the success of reinforcement learning from human feedback (RLHF) (Christiano et al., 2017) on state-of-the-art and high-profile language models such as GPT-4 (OpenAI, 2024). The goal of learning from preference data is to finetune powerful base language models to output generations more in line with human preferences (Stiennon et al., 2020; Ouyang et al., 2022), motivated by the fact that pretraining on internet-scale data has enabled large language models to exhibit fluent generations of text (Minaee et al., 2024) but not necessarily responses aligned with what humans prefer to see.

In recent years, the landscape of algorithms and evaluations for preference learning has grown significantly (Kaufmann et al., 2023; Jiang et al., 2024), resulting in a complex and often fragmented field that can pose challenges for practitioners and researchers deciding where to focus their efforts. Whereas well-established machine learning tasks such as classification are grounded in principles such as maximum likelihood, preference learning lacks such a unifying conceptual foundation. To advance the field, we thus first ask: what underlying principles underpin preference learning?

We address this question by developing a framework starting from the sampling distribution implied by pairwise preference data. We first tackle the question of evaluating a model under the preference learning paradigm. We show that the only evaluation of a generative model rooted in the preference data sampling distribution itself is a form of win rate we call $h$-win rate; any other notions of good either do not evaluate the model or respect the preference data, or are based on assumptions outside of either (Section 3). From this insight, we introduce a win rate-centric framework for understanding the landscape of preference learning methods. Given $h$-win rate is the only relevant evaluation without additional assumptions, we relate common preference learning algorithms to directly optimizing for win rate, dividing the preference learning space into win rate optimization (WRO) and non-WRO objectives. In Section 4, we generalize the space of WRO beyond existing methods in the literature and identify two theoretical benefits of such methods: (1) win rate-correspondence, i.e., optimizing the objective corresponds to optimizing for $h$-win rate; and (2) win rate-consistency, i.e., the solution can achieve the maximum win rate possible over a competitor

[1]Center for Data Science, New York University, New York, USA [2]Courant Institute, New York University, New York, USA. Correspondence to: Lily H. Zhang <lily.h.zhang@nyu.edu>.

*Proceedings of the 42$^{nd}$ International Conference on Machine Learning*, Vancouver, Canada. PMLR 267, 2025. Copyright 2025 by the author(s).

as regularization strength goes to zero. In Section 5, we show that direct preference optimization (DPO) and other direct alignment algorithms (Rafailov et al., 2024a) fail win rate-correspondence, and that supervised finetuning (SFT) on preferred samples additionally fails win rate-consistency. In Section 6, we show that despite their theoretical benefits, WRO methods underperform relative to expectations due to difficulties in optimization, which has a much more significant effect on performance than other design choices that distinguish WRO methods from each other. We conclude with takeaways for current practice and future research in preference learning (Section 7).

Our contributions can be summarized as follows:

1. We prove that the only evaluation of a generative model grounded in the preference data distribution is win rate. This result justifies using win rate as the focal point to understand the landscape of preference learning.

2. We present a win rate-centric framework to understand preference learning. Under this framework, we:

   (a) introduce win rate-correspondence and consistency and show that WRO methods satisfy both while non-WRO do not;

   (b) highlight the central role of optimization success for the performance;

   (c) motivate best practices for existing methods (e.g., win rate checkpointing with direct alignment algorithms, multiple seeds with WRO algorithms, inducing diversity for SFT, combining different approaches); and

   (d) provide recommendations for future research—namely, to focus on better surrogates for or improved optimization of WRO.

## 2. Related Work

Our work is most closely related to previous work in win rate evaluation and optimization as well as analysis of RLHF and preference learning objectives.

**Win rate evaluation and optimization.** Win rate is already a central evaluation in preference learning (Li et al., 2023b; Zheng et al., 2024); however, our work goes further and proves that it is the only evaluation grounded in the sampling distribution itself, thus motivating its use as the central object to understand the rest of the preference learning landscape. Several works have proposed methods that perform some form of win rate optimization (Munos et al., 2023; Swamy et al., 2024; Rosset et al., 2024), including as a minimax optimization with a dynamic (rather than fixed) opponent model. Our work provides a more general framework to characterize the landscape of win rate optimization

(WRO) methods beyond the specific instances that currently exist in the literature. Moreover, our analysis points to the theoretical merit of these approaches, while our experiments demonstrate the implementation bottlenecks of them.

**Analyzing RLHF, DPO, and other preference learning methods.** Our work is related to work that seeks to better understand RLHF, DPO, and other existing methods in preference learning (e.g., best-of-n). Examples include benchmarking generalization and diversity (Kirk et al., 2024), comparing on- vs off-policy approaches (Tajwar et al., 2024), investigating length bias (Singhal et al., 2024), and disentangling design choices empirically (Ivison et al., 2024). For RLHF, existing works consider the complexity of proximal policy optimization (Ahmadian et al., 2024), vanishing gradients (Razin et al., 2024b), reward model overoptimization (Zhu et al., 2024), or limitations of the Bradley-Terry assumption to relate preferences to rewards (Wang et al., 2024b; Azar et al., 2023; Munos et al., 2023; Siththaranjan et al., 2024). For DPO, there exists not only large space of alternative direct alignment algorithms (e.g., (Zhao et al., 2023; Azar et al., 2023; Xu et al., 2024a; Huang et al., 2024; Pal et al., 2024; Xu et al., 2024b)) but also works which analyze its limited ability to flip rankings (Chen et al., 2024), the decrease in chosen and rejected log probabilities (Razin et al., 2024a), and its limitations under heterogeneous preferences (Shirali et al., 2025). Gui et al. (2024) show that the target distribution of best-of-n for different choices of $n$ yields a win rate vs. KL divergence curve close to that of RLHF's for different choices of divergence constraint; Theorem 5.1 of this work generalizes their win rate result for best-of-2 from a deterministic to an arbitrary preference environment. Azar et al. (2023) present a family of $\Psi$-Preference Optimization objectives, show that RLHF falls within this family, and propose Identity Preference Optimization (IPO); this work presents a larger family of WRO objectives beyond $\Psi$-Preference Optimization and categorizes $\Psi$-Preference Optimization and IPO separately into WRO and non-WRO objectives, highlighting theoretical benefits of the former and limitations of the latter.

## 3. What evaluations make sense for preference learning?

In this section, we analyze how to evaluate a generative model under the preference learning landscape. Various evaluations have been used in the preference learning literature, from win rate (Li et al., 2023b) and reward (Ahmadian et al., 2024) to implicit reward accuracy (Rafailov et al., 2024a) and ranking accuracy (Chen et al., 2024). Here, we critically assess what evaluations are meaningful under the pairwise preference comparisons data itself.

We first describe the underlying sampling distribution of the pariwise comparison data used in preference learning (Sec-

tion 3.1). Then we define and motivate two important conditions for an evaluation function, preference-consistency and prevalence-consistency, and prove that the only evaluations that satisfy both conditions is a form of win rate we call $h$-win rate (Section 3.2). This result justifies win rate as the singular focus in preference learning.

### 3.1. The Sampling Distribution

The goal of preference learning is to learn a generative model that performs well in a given context. However, whereas typical maximum likelihood training employs samples from the distribution of interest, the setup of preference learning does not: only samples from generative model competitors (sometimes the same model) and their relative preference within a pair are available.

The sampling distribution for preferences consists of input $\mathbf{x}$, candidate outputs $\mathbf{y}_0$ and $\mathbf{y}_1$, and a label $\ell \in \{0, 1\}$ denoting which of $\mathbf{y}_0$ or $\mathbf{y}_1$ is preferred. Let $\ell = 1$ denote that $\mathbf{y}_1$ is preferred, and $\ell = 0$ denote that $\mathbf{y}_0$ is preferred. Then, the overall sampling distribution can be defined as follows:

**Definition 3.1.** A **sampling distribution for (pairwise) preference learning** is a distribution over input $\mathbf{x} \in \mathcal{X}$, candidate outputs $\mathbf{y}_0, \mathbf{y}_1 \in \mathcal{Y}$, and preference label $\ell \in \{0, 1\}$ defined by:

1. Query distribution: $p(\mathbf{x})$

2. Generation competitor 0: $p(\mathbf{y}_0 \mid \mathbf{x})$

3. Generation competitor 1: $p(\mathbf{y}_1 \mid \mathbf{x})$

4. Preference classifier: $p(\ell \mid \mathbf{x}, \mathbf{y}_0, \mathbf{y}_1)$.

1, 2, and 3 are *user-specified* distributions; 1 denotes the inputs of interest, and 2 and 3 are the candidate competitors one chooses to evaluate. 4 is only distribution that cannot be directly specified; rather, it is defined by the environment in which the user chooses to collect the preferences.

Generation competitor 0 and 1 can be the same distribution and often are in existing open-source preference datasets (Lambert et al., 2023; Ethayarajh et al., 2022).

### 3.2. Win Rate is the only evaluation that can matter

The goal of preference learning is to learn some generative model $p^*(\mathbf{y} \mid \mathbf{x})$ that performs well under the preference environment for a given query distribution (we refer to this as the *query-preference environment*). Learning such a generative model requires a definition of what is good.

Consider an evaluation function $\phi$ which maps a generative model $p(\mathbf{y} \mid \mathbf{x})$, query-preference environment $\mathcal{E} = (p(\mathbf{x}), p(\ell \mid \mathbf{x}, \mathbf{y}_0, \mathbf{y}_1))$, and anchor distribution $p(\mathbf{y}_0 \mid \mathbf{x})$ to

a scalar: $\phi_{p(\mathbf{y}_0 \mid \mathbf{x})}(p(\mathbf{y} \mid \mathbf{x}), \mathcal{E}) \in \mathbb{R}$.[1] Intuitively, $\phi$ should respect properties of the preference sampling distribution. We formalize criterion for $\phi$ below.

**Definition 3.2.** Any evaluation function $\phi$ is **grounded** in a given preference distribution if

1. ($\phi$ is preference-consistent): given a strictly increasing function $h$, then for any singleton query environment $p(\mathbf{x}) = \mathbb{1}[\mathbf{x} = \mathbf{x}']$, generative model $p(\mathbf{y} \mid \mathbf{x}) = \mathbb{1}[\mathbf{y} = \mathbf{y}']$, anchor distribution $p(\mathbf{y}_0 \mid \mathbf{x}) = \mathbb{1}[\mathbf{y}_0 = \mathbf{y}_0']$:

$$\phi(p(\mathbf{y} \mid \mathbf{x}), \mathcal{E}) = h \cdot p(\ell = 1 \mid \mathbf{x}', \mathbf{y}_0', \mathbf{y}'); \text{ and}$$

2. ($\phi$ is prevalence-consistent): for $a, b \geq 0$ and $a + b = 1$:

   (a) for generator $p(\mathbf{y} \mid \mathbf{x}) = ap_1(\mathbf{y} \mid \mathbf{x}) + bp_2(\mathbf{y} \mid \mathbf{x})$:

   $$\phi(p(\mathbf{y} \mid \mathbf{x}), \mathcal{E}) = a\phi(p_1(\mathbf{y} \mid \mathbf{x}), \mathcal{E}) + b\phi(p_2(\mathbf{y} \mid \mathbf{x}), \mathcal{E});$$

   (b) for query distribution $p(\mathbf{x}) = ap_1(\mathbf{x}) + bp_2(\mathbf{x})$, letting $\mathcal{E}_i = (p_i(\mathbf{x}), p(\ell = 1 \mid \mathbf{x}, \mathbf{y}_0, \mathbf{y}_1))$:

   $$\phi(p(\mathbf{y} \mid \mathbf{x}), \mathcal{E}) = a\phi(p(\mathbf{y} \mid \mathbf{x}), \mathcal{E}_1) + b\phi(p(\mathbf{y} \mid \mathbf{x}), \mathcal{E}_2);$$

   (c) for anchor distribution $p(\mathbf{y}_0 \mid \mathbf{x}) = ap_1(\mathbf{y}_0 \mid \mathbf{x}) + bp_2(\mathbf{y}_0 \mid \mathbf{x})$ :

   $$\phi_{p(\mathbf{y}_0 \mid \mathbf{x})}(p(\mathbf{y} \mid \mathbf{x}), \mathcal{E}) = a\phi_{p_1(\mathbf{y}_0 \mid \mathbf{x})}(p(\mathbf{y} \mid \mathbf{x}), \mathcal{E})$$
   $$+ b\phi_{p_2(\mathbf{y}_0 \mid \mathbf{x})}(p(\mathbf{y} \mid \mathbf{x}), \mathcal{E}).$$

Preference-consistency ensures that the evaluation can be reduced to an increasing function of the preference classifier in the base case where query, model, and anchor distributions are all a single deterministic value. The transformation function $h$ dictates that the evaluation need not exactly equal the preference probability, just that the evaluation should be higher when the preference probability is higher. Prevalence-consistency ensures that the contribution of $p(\ell = 1 \mid \mathbf{x}, \mathbf{y}_0, \mathbf{y}_1)$ for a given $(\mathbf{x}, \mathbf{y}_0, \mathbf{y}_1)$ gets weighted appropriately by its prevalence in the query environment, anchor distribution, and generative model, respectively.

What are the consequences if these properties are not met? If preference-consistency is not satisfied, a model can be assigned a high value under $\phi$ even if it only outputs a dispreferred response according to the ground truth preference. For prevalence-consistency, if 2a is not satisfied, then $\phi$ weights preferences according to a distribution that is *not* how the model generates; if 2b is not satisfied, then $\phi$ does not reflect the chosen queries; and if 2c is not satisfied, then $\phi$ does not compare to the chosen anchor.

The only evaluation that satisfies Definition 3.2 is a form of win rate:

---

[1] We will optionally write $\phi_{p(\mathbf{y}_0 \mid \mathbf{x})}$ as $\phi$ when the anchor is clear from context.

**Proposition 3.3.** *For $p$ with discrete support, $\phi$ is grounded, as defined in Definition 3.2, if and only if*

$$\phi_{p(\boldsymbol{y}_0 \,|\, \boldsymbol{x})}(p(\boldsymbol{y} \,|\, \boldsymbol{x}), \mathcal{E})$$
$$=\mathbb{E}_{p(\boldsymbol{x})}\mathbb{E}_{p(\boldsymbol{y} \,|\, \boldsymbol{x})}\mathbb{E}_{p(\boldsymbol{y}_0 \,|\, \boldsymbol{x})}[h \cdot p(\ell = 1 \,|\, \boldsymbol{x}, \boldsymbol{y}_0, \boldsymbol{y})] \quad (1)$$

*for choice of strictly increasing function h.*

See Appendix B for proof. Equation (1) is the win rate of the generator $p(\mathbf{y} \,|\, \mathbf{x})$ against an anchor distribution $p(\mathbf{y}_0 \,|\, \mathbf{x})$ under the query-preference environment $\mathcal{E}$, for some choice of order-preserving transformation function $h$. When $h$ is the identity, we have vanilla win rate, which we will refer to as win rate or Win Rate$_{p(\mathbf{y}_0 \,|\, \mathbf{x})}[p(\mathbf{y} \,|\, \mathbf{x})]$. Often it is estimated via samples from the preference classifier, i.e., $\mathbb{E}_{p(\mathbf{x}, \mathbf{y}, \mathbf{y}_0)}\ell$, where $\ell \sim p(\ell \,|\, \mathbf{x}, \mathbf{y}_0, \mathbf{y})$ (Li et al., 2023b). When $h$ is any other strictly increasing function, we have an $h$-variant of win rate, which we will refer to as $h$-Win Rate.

Different $h$-variants can stretch preference probabilities in some regions while contracting them in others, effectively prioritizing certain preference probability differences over others in the evaluation of generator vs. anchor. Note that placing non-identity functions $f$, $g$ in any other position, i.e., $\mathbb{E}_{p(\mathbf{x})}f \cdot \mathbb{E}_{p(\mathbf{y} \,|\, \mathbf{x})}g \cdot \mathbb{E}_{p(\mathbf{y}_0 \,|\, \mathbf{x})}[h \cdot p(\ell = 1 \,|\, \mathbf{x}, \mathbf{y}_0, \mathbf{y})]$, breaks prevalence-consistency of query and generator respectively.

What about other evaluations? Proposition 3.3 proves that all other evaluations either break preference- or prevalance-consistency. For instance, reward and ranking accuracy, typically computed on offline data, do not respect model prevalences given the pairs in the computation may be unlikely overall under the model; empirically, they have also been shown to correlate little with win rate (Rafailov et al., 2024a). Average reward $\mathbb{E}_{p(\mathbf{x}, \mathbf{y})}r(\mathbf{x}, \mathbf{y})$ is equivalent to a translated logit-Win Rate under the Bradley-Terry assumption $p(\ell = 1 \,|\, \mathbf{x}, \mathbf{y}_0, \mathbf{y}) = \sigma[r(\mathbf{x}, \mathbf{y}) - r(\mathbf{x}, \mathbf{y}_0)]$ (Azar et al., 2023), but the underlying preference environment need not satisfy this assumption.[2]

Given the insight that the only evaluation grounded in preference data alone is $h$-win rate with respect to some anchor distribution, we next analyze common preference learning algorithms based on how they relate to optimizing for win rate. In Section 4, we characterize the space of Win Rate Optimization (WRO) objectives, highlighting existing and new methods that fall under this family, and discuss their theoretical benefits. In Section 5, we analyze instances of non-WRO methods and their limitations.

# 4. Preference learning through the Lens of Win Rate Optimization

In this section, we characterize the space of objectives which directly optimize for win rate, discussing both Win Rate Optimization (WRO) objectives (Section 4.1) and games (Section 4.2). Then, we identify the theoretical benefits shared by all methods in the WRO family (Section 4.3).

## 4.1. Win Rate Optimization Objectives

The fact that Equation (1) is the only grounded evaluation immediately provides an objective $\mathcal{O}_h(\theta)$ to optimize:

$$\max_{\theta} \mathcal{O}_h(\theta) = \max_{\theta} h\text{-Win Rate}_{p(\mathbf{y}_0 \,|\, \mathbf{x})}[p_{\theta}(\mathbf{y}_1 \,|\, \mathbf{x})]$$
$$= \max_{\theta} \mathbb{E}_{p(\mathbf{x})p_{\theta}(\mathbf{y}_1 \,|\, \mathbf{x})p(\mathbf{y}_0 \,|\, \mathbf{x})}[h \cdot p(\ell = 1 \,|\, \mathbf{x}, \mathbf{y}_0, \mathbf{y}_1)]. \quad (2)$$

WRO can be optimized by using score function/policy gradients (Mohamed et al., 2020a; Weng, 2018). The solution to any objective in this family is the generator that maximizes the $h$-win rate over a given anchor.

KL-constrained RLHF is KL-regularized WRO with $h$=logit and the Bradley-Terry assumption to estimate the preference classifier via a reward function (Azar et al., 2023). However, many other WRO instances exist, e.g., for any combination of $h$, anchor distribution, and estimate of $p(\ell = 1 \,|\, \mathbf{x}, \mathbf{y}_0, \mathbf{y}_1)$ or $\mathbb{E}_{p(\mathbf{y}_0 \,|\, \mathbf{x})}[h \cdot p(\ell = 1 \,|\, \mathbf{x}, \mathbf{y}_0, \mathbf{y}_1)]$ yields a different WRO objective.[3] Different choices of $h$ and anchor yield different optimal solutions in general. When the BT assumption holds, however, all objectives in Equation (2) maximize all $h$-variants of win rate for all anchors.

**Proposition 4.1.** *(informal) Under the Bradley-Terry assumption, all objectives in Equation (2) with strictly increasing $h$ share the same optimal solution. This solution maximizes win rate overall all possible anchor distributions.*

See Appendix C for proof.

## 4.2. Win Rate Optimization Games

The choice of anchor need not be a fixed up front. Instead, the anchor distribution could also optimize for its own win

---

[2]Note that the Bradley Terry assumption suggests not only transitivity but a functional relationship in the preference probabilities, e.g., if $A$ is preferred over $B$ with probability .6 and $B$ is preferred over $C$ with probability .7, then the Bradley-Terry assumption states that $A$ must be preferred over $C$ with probability $\approx 0.778$.

[3]For instance, $\mathbb{E}_{p(\mathbf{y}_0 \,|\, \mathbf{x})}[h \cdot p(\ell = 1 \,|\, \mathbf{x}, \mathbf{y}_0, \mathbf{y}_1)]$ can estimated by learning the preference distribution $p(\ell = 1 \,|\, \mathbf{x}, \mathbf{y}_0, \mathbf{y}_1)$ from data and estimating the expectation with samples from the anchor or training a model to estimate this expectation directly, removing the need to additionally sample from $p(\mathbf{y}_0 \,|\, \mathbf{x})$ during the policy optimization step. The RLHF setting of WRO in particular (namely, choice of $h$ = logit and BT assumption) also allows the expectation over the anchor $p(\mathbf{y}_0 \,|\, \mathbf{x})$ to be dropped, i.e., $\max_{p(\mathbf{y}_1 \,|\, \mathbf{x})} \mathbb{E}_{p(\mathbf{y}_1 \,|\, \mathbf{x})}\mathbb{E}_{p(\mathbf{y}_0 \,|\, \mathbf{x})}[h \cdot p(\ell = 1 \,|\, \mathbf{x}, \mathbf{y}_0, \mathbf{y}_1)] = \max_{p(\mathbf{y}_1 \,|\, \mathbf{x})} \mathbb{E}_{p(\mathbf{y}_1 \,|\, \mathbf{x})}[r(\mathbf{x}, \mathbf{y})]$, thus requiring only that a reward model be estimated.

rate, resulting in a family of WRO games:

$$\max_{\theta} \mathbb{E}_{p(\mathbf{x})}\mathbb{E}_{p_\theta(\mathbf{y}|\mathbf{x})}\mathbb{E}_{p_\gamma(\mathbf{y}'|\mathbf{x})}[h \cdot p(\ell = 1|\mathbf{x}, \mathbf{y}', \mathbf{y})],$$

$$\max_{\gamma} \mathbb{E}_{p(\mathbf{x})}\mathbb{E}_{p_\theta(\mathbf{y}|\mathbf{x})}\mathbb{E}_{p_\gamma(\mathbf{y}'|\mathbf{x})}[h \cdot p(\ell = 1|\mathbf{x}, \mathbf{y}, \mathbf{y}')].$$

Nash Learning from Human Feedback (Munos et al., 2023) is one instance of a WRO game with $h$=identity (and additional KL regularization); the choice of $h$ leads to an antisymmetric, constant-sum game. However, WRO games need not be anti-symmetric or constant sum; for instance, a WRO game with $h = \log$ is neither. Different choices of $h$ dictate a different prioritization of preference probabilities to target, and it is not obvious that $h$=identity, the specific setting considered in Munos et al. (2023), should be the only WRO game to consider. After all, aligning to preferences is not inherently a constant sum endeavor.

### 4.3. Benefits of Win Rate Optimization

Letting $\phi_h(\theta)$ denote the $h$-win rate, $\phi_h(p_\theta(\mathbf{y}|\mathbf{x}))$, two benefits of WRO objectives $\mathcal{O}_h(\theta)$ are:

1. (Win rate-correspondence): Improving the objective improves $h$-win rate:

$$\mathcal{O}(\theta) > \mathcal{O}(\theta') \Rightarrow \phi(\theta) > \phi(\theta').$$

2. (Win rate-consistency): The optimum of the objective maximizes $h$-win rate:

$$\text{For } \theta^* = \arg\max_{\theta} \mathcal{O}(\theta) : \phi(\theta^*) = \max_{\theta} \phi(\theta).$$

Win rate-correspondence makes gradient-based optimization practical, while win rate-consistency ensures that the loss itself does not exhibit fundamental limitations with respect to its optimal solution.[4] Both properties follow trivially for WRO from the fact that $\mathcal{O} = \phi$. These properties need not hold for non-WRO (see Section 5).

**Regularization.** As $p(\ell = 1 | \mathbf{x}, \mathbf{y}_0, \mathbf{y}_1)$ is typically estimated from data, to prevent overfitting to it one can regularize with with a divergence penalty $D(p_\theta, p_{\text{ref}})$ with regularization parameter $\beta$:

$$\max_{\theta} h\text{-Win Rate}_{p(\mathbf{y}_0|\mathbf{x})}[p_\theta(\mathbf{y}_1|\mathbf{x})] - \beta\mathbb{E}_{p(\mathbf{x})}D(p_\theta, p_{\text{ref}}).$$

Options for $D$ include sequence-level reverse KL, chi-sq, or a sum of token-level divergences (Azar et al., 2023; Huang et al., 2024; Rafailov et al., 2024b).

Optimizing WRO with reverse KL regularization is equivalent to minimizing the reverse KL divergence between

the model and the target distribution $p^*_{\text{WRO-KL}}(\mathbf{y}|\mathbf{x}) \propto p_{\text{ref}}(\mathbf{y}|\mathbf{x})\exp(\frac{1}{\beta}\mathbb{E}_{p(\mathbf{y}_0|\mathbf{x})}[h \cdot p(\ell = 1|\mathbf{x}, \mathbf{y}_0, \mathbf{y})])$; see Azar et al. (2023) or Appendix D.1 for derivation. As such, the reverse-KL regularized WRO objective is a form of black-box variational inference (Ranganath et al., 2014).

Regularized-WRO $\mathcal{O}_{h,\beta}(\theta)$ does not satisfy win rate-correspondence nor consistency in general but modified versions of both: 1. (regularized win rate-correspondence): Improving the objective implies either an improvement in $h$-win rate or divergence to the reference, i.e., the regularization term; and 2. (regularized win rate-consistency): $\theta^* = \arg\max_{\theta} \mathcal{O}_{h,\beta}(\theta)$ maximizes $h$-win rate for all distributions with divergence within $M(\beta) = D(p_{\theta^*}, p_{\text{ref}})$.

## 5. Preference algorithms that are not WRO

### 5.1. DPO fails win rate-correspondence

Direct Preference Optimization (DPO) (Rafailov et al., 2024c) shares the same target distribution as RLHF but employs a different objective to optimize for that target. Denoting $p_\theta(\ell | \mathbf{x}, \mathbf{y}_0, \mathbf{y}_1)$ as $p_{\ell,\theta}$ and $q(\mathbf{x}, \mathbf{y}_0, \mathbf{y}_1)$ as some distribution of offline data, DPO is

$$\min_{\theta} \mathcal{L}_{\text{DPO}}(\theta) = \min_{\theta} -\mathbb{E}_{q(\mathbf{x},\mathbf{y}_0,\mathbf{y}_1)p(\ell|\mathbf{x},\mathbf{y}_0,\mathbf{y}_1)}[\log p_{\ell,\theta}], \quad (3)$$

where $p_{\ell,\theta}$ is parametrized to include a language model inside of it. Namely, $p_{\ell,\theta}(\ell = 1 | \mathbf{x}, \mathbf{y}_0, \mathbf{y}_1)$ is

$$\sigma\left[\beta\log\frac{p_\theta(\mathbf{y}_1|\mathbf{x})}{p_{\text{ref}}(\mathbf{y}_1|\mathbf{x})} - \beta\log\frac{p_\theta(\mathbf{y}_0|\mathbf{x})}{p_{\text{ref}}(\mathbf{y}_0|\mathbf{x})}\right]. \quad (4)$$

DPO fails win rate-correspondence, as better (i.e., lower) DPO loss does not necessarily mean better (i.e., higher) win rate. This correspondence fails for two reasons:

1. Using offline data breaks prevalence-consistency. For instance, a model change that improves the ranking of responses $y_A, y_B$ in the offline data but moves more mass overall to generate dispreferred responses not covered in the offline data will improve DPO loss but hurt win rate. Even sampling from the initial model is not sufficient (see Appendix F for an example), as prevalence-consistency of the generator does not hold throughout training.

2. Even if we satisfy prevalence-consistency by switching to a fully online setting[5], win rate correspondence still does not hold; see Figure 1 for a visualization. This is because the likelihood of preference

---

[4]For readers interested in the precedence for these properties, win rate-correspondence and consistency are closely linked to Question 2 in (Steinwart, 2007) and $\mathcal{H}$-consistency (Awasthi et al., 2021) respectively.

[5]This would consist of sampling from the appropriate query, generator, and anchor distributions throughout training. Guo et al. (2024); Calandriello et al. (2024) do this but use $\mathbb{E}_{\text{stop-gradient}(p_\theta(\mathbf{y}|\mathbf{x}))}$ instead of $\mathbb{E}_{p_\theta(\mathbf{y}|\mathbf{x})}$ as a loss; as evaluations both are the same.

distribution as in Equation (3) is not an $h$-win rate: $-\mathbb{E}_{p_\theta(\mathbf{x},\mathbf{y}_0,\mathbf{y}_1)}[\mathbb{E}_{p_\ell} \log p_{\ell,\theta}] \neq \mathbb{E}_{p_\theta(\mathbf{x},\mathbf{y}_0,\mathbf{y}_1)}[h \cdot p_\ell]$. Since prevalence-consistency holds, preference consistency of this online objective must not hold, which can be seen by noting that the argument inside $\mathbb{E}_{p_\theta(\mathbf{x},\mathbf{y}_0,\mathbf{y}_1)}$ is not a fixed function of $p_\ell$.

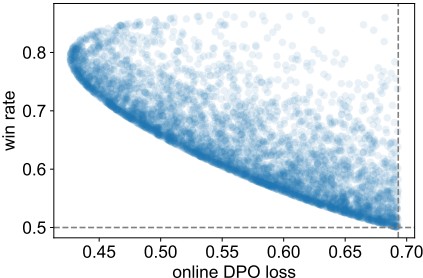

Figure 1: Plot of win rate vs. DPO loss for different $p_\theta(\mathbf{y}\,|\,\mathbf{x})$ for a given choice of anchor and preference classifier (see Appendix F.2 for details). Any pair of points that would form a line segment with a positive slope form a setting of $\theta, \theta'$ such that win rate correspondence does not hold. Grey lines denote the win rate and DPO loss at initialization. See Appendix F.2 for additional discussion and examples.

Note that even though DPO is derived to match the target distribution of a regularized-WRO objective, DPO does not satisfy regularized win rate-correspondence either. Namely, it is possible to improve (i.e., decrease) the DPO loss without improving win rate or divergence to the reference model (see Appendix F.3 for example).

The insight that DPO fails win rate-correspondence offers an explanation for empirical results such as the loss vs. win rate misalignment observed in Chen et al. (2024); Rafailov et al. (2024a), as well as for why it can helpful to perform very early stopping (e.g., one epoch, far from any evidence of overfitting) rather than model selection with DPO validation loss (Rafailov et al., 2024c), or to choose checkpoints using win rate directly (Yuan et al., 2024).

The above also applies to direct alignment algorithms (DAA) in general (Azar et al., 2023; Tang et al., 2024; Huang et al., 2024; Ji et al., 2024): the use of offline data means that prevalence-consistency does not hold, and an inner objective $f(\mathbf{x},\mathbf{y}_0,\mathbf{y}_1)$ in $\mathbb{E}_{q(\mathbf{x},\mathbf{y}_0,\mathbf{y}_1)}f(\mathbf{x},\mathbf{y}_0,\mathbf{y}_1)$ other than the preference classifier (including divergence to the true classifier, distance to some function of the true classifier) means that preference-consistency does not hold.

**Takeaways.** Even though DPO and other DAA often share the same optimal solution as a regularized-WRO objective (i.e., regularized-win rate consistency holds), these alternative objectives fail win rate-correspondence, complicating their use in training and model selection.

## 5.2. SFT fails win rate-correspondence and consistency

Supervised finetuning (SFT) is often viewed as an initial step for or necessary precursor to other preference learning algorithms (Wang et al., 2024a; Razin et al., 2024b), but it is also a preference learning algorithm itself based on filtering and maximum likelihood estimation.

Supervised finetuning on preferred samples (sometimes denoted $\mathbf{y}_w$) seeks to maximize the likelihood of sample $\mathbf{y}_1$ when $\ell = 1$ and $\mathbf{y}_0$ when $\ell = 0$. When $p(\mathbf{y}_1\,|\,\mathbf{x}) = p(\mathbf{y}_0\,|\,\mathbf{x})$, then $p(\mathbf{y}_w\,|\,\mathbf{x}) = p(\mathbf{y}_1\,|\,\mathbf{x}, \ell = 1) = p(\mathbf{y}_0\,|\,\mathbf{x}, \ell = 0)$ and the SFT loss can be written as follows:

$$\mathcal{L}_{\text{SFT}}(\theta) = \min_\theta -\mathbb{E}_{p(\mathbf{x},\mathbf{y}_1|\ell)} \log p_\theta(\mathbf{y}_1\,|\,\mathbf{x}).$$

Like DPO and direct alignment algorithms, this fails win rate-correspondence, as the above expression is not an $h$-win rate. What about win rate consistency? The target distribution can be written as

$$p^*_{\text{SFT}}(\mathbf{y}\,|\,\mathbf{x}) \propto p(\mathbf{y}\,|\,\mathbf{x})\mathbb{E}_{p(\mathbf{y}_0\,|\,\mathbf{x})}[p(\ell = 1\,|\,\mathbf{x},\mathbf{y}_0,\mathbf{y})], \quad (5)$$

which tilts the original distribution towards sequences with higher average preference probabilities over the anchor. However, there are limits to the amount of improvement Equation (5) can achieve (see Appendix E for visualization). In fact, we can characterize the exact win rate expected from the SFT target distribution over the original model:

**Theorem 5.1** (Win rate improvement of SFT on preferred). *Let $p(\mathbf{y}_0\,|\,\mathbf{x})$ be the initial generative model, and $p_{SFT}(\mathbf{y}\,|\,\mathbf{x})$ be the target distribution of SFT on preferred samples $(p(\mathbf{y}_1\,|\,\mathbf{x}, \ell = 1), p(\mathbf{y}_0\,|\,\mathbf{x}) = p(\mathbf{y}_1\,|\,\mathbf{x}))$. Then,*

$$\begin{aligned} Win\ Rate_{p(\mathbf{y}_0\,|\,\mathbf{x})}[p_{SFT}(\mathbf{y}\,|\,\mathbf{x})] &= 0.5 \\ + 2\mathbb{E}_{p(\mathbf{x})}Var_{p(\mathbf{y}_1\,|\,\mathbf{x})}&\left[\mathbb{E}_{p(\mathbf{y}_0\,|\,\mathbf{x})}[p(\ell = 1\,|\,\mathbf{x},\mathbf{y}_0,\mathbf{y}_1)]\right]. \quad (6) \end{aligned}$$

The theorem states that variance in the average preference probability under the initial model dictates the extent of win rate improvement. Intuitively, one should not expect any possible improvement in win rate if the existing model only outputs sequences which are equally preferred to each other; on the other hand, there is more room for improvement in win rate the more differentially preferred some of the model's sequences are to others. As long as the model has support over more than two sequences, however, the win rate cannot be maximized. See Appendix G for proof.

We can generalize Theorem 5.1 to other filtering strategies:

**Theorem 5.2** (Win rate improvement of filter + SFT). *Let $p(f\,|\,\mathbf{x},\mathbf{y}_1,\mathbf{y}_0,\ell)$ be a filter that selects data ($f = 1$) for supervised finetuning, and $p(\mathbf{y}_0\,|\,\mathbf{x}) = p(\mathbf{y}_1\,|\,\mathbf{x})$). Then, denoting $\mathbb{E}_{p(\ell,\mathbf{y}_0\,|\,\mathbf{x})}[p(f = 1\,|\,\mathbf{x},\mathbf{y}_0,\mathbf{y}_1,\ell)]$ as AvgFilter$(\mathbf{x},\mathbf{y}_1)$*

*and* $\mathbb{E}_{p(\mathbf{y}_0 \,|\, \mathbf{x})}\left[p(\ell = 1 \,|\, \mathbf{x}, \mathbf{y}_0, \mathbf{y}_1)\right]$ *as* $AvgPref(\mathbf{x}, \mathbf{y}_1)$,

$$Win\ Rate_{p(\mathbf{y}_0 \,|\, \mathbf{x})}[p_{SFT}(\mathbf{y} \,|\, \mathbf{x})] = 0.5$$
$$+ \mathbb{E}_{p(\mathbf{x})} \frac{Cov_{p(\mathbf{y}_1 \,|\, \mathbf{x})}[AvgFilter(\mathbf{x}, \mathbf{y}_1), AvgPref(\mathbf{x}, \mathbf{y}_1)]}{\mathbb{E}_{p(\mathbf{y}_1 \,|\, \mathbf{x})} AvgFilter(\mathbf{x}, \mathbf{y}_1)}. \quad (7)$$

See Appendix G for proof. Doing SFT on preferred sets the filter as $p(f = 1 \,|\, \mathbf{x}, \mathbf{y}_1, \mathbf{y}_0, \ell) = \mathbf{1}[\ell = 1]$. Win rate increases as the denominator of the second term, measuring how often a response is selected, decreases and/or the numerator, measuring how closely the average filter and preference co-vary increases. In other words, both the threshold for filtering as well as the diversity (in preference)[6] of the initial model matter for improvement possible in a filter + SFT algorithm. This could explain the success of works such as Li et al. (2023a), who use backtranslation to generate a larger set of responses than the model's original generations and filter a subset for SFT, and Brandfonbrener et al. (2021), who filter based on quantile.

**Takeaways.** SFT on preferred samples fails win rate-correspondence and consistency. The improvement possible is a function of the initial model's diversity in preference. Other filtering strategies with stricter criteria can yield further win rate improvements.

# 6. Investigating the empirical impact of different design choices

We next compare WRO and non-WRO experimentally to complement the above theoretical analysis with an empirical one. First, we compare SFT, DPO, and RLHF on equal footing based on how much they are able to improve win rate over the starting model. Moreover, we also compare different variants of KL-regularized WRO, varying $h$ (identity, log, and logit), $\beta$ (1, 0.1, 0.01, 0.001), and the estimation of the preference classifier (perfect and estimated with and without the Bradley Terry assumption).

## 6.1. Experimental setup

We employ Pythia-2.8b (Biderman et al., 2023) as our base model and the OpenAssistant (OASST) (Köpf et al., 2023) and Anthropic Helpfulness and Harmlessness (HH) (Bai et al., 2022) as datasets. We train the base model on the data outputs and use these finetuned models as our initial models. We train an oracle judge model per dataset to estimate $p(\ell = 1 \,|\, \mathbf{x}, \mathbf{y}_0, \mathbf{y}_1)$ and relabel the preference annotations using this judge model. We use the same oracle to evaluate win rate after training. See Appendix K for further details on the judge model. We additionally train 1. a reward model on the oracle-labeled judge annotations (accuracy is 82.8

---

[6]Recall that $Cov_{p(\mathbf{y}_1 \,|\, \mathbf{x})}[AvgFilter(\mathbf{x}, \mathbf{y}_1), AvgPref(\mathbf{x}, \mathbf{y}_1)] \leq \sqrt{Var_{p(\mathbf{y}_1 \,|\, \mathbf{x})}[AvgFilter(\mathbf{x}, \mathbf{y}_1)] Var_{p(\mathbf{y}_1 \,|\, \mathbf{x})}[AvgPref(\mathbf{x}, \mathbf{y}_1)]}$.

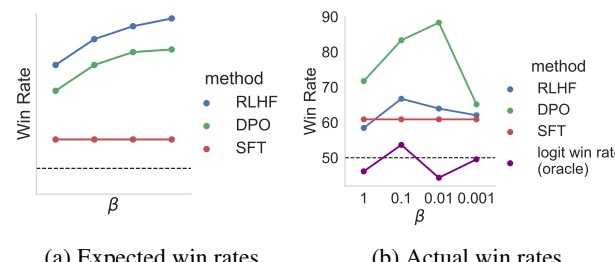

(a) Expected win rates        (b) Actual win rates

Figure 2: Expected versus observed win rates of RLHF, DPO, and SFT over the original model for OASST. RLHF notably underperforms relative to expectations. Moreover, substituting the learned reward model for the oracle preference classifier (logit win rate (oracle)) does not improve performance, suggesting other factors are more important.

for OASST and 81.36 for HH) and 2. an imperfect judge model (accuracy is 80.47 for OASST and 85.16 for HH). As expected, the BT assumption is helpful for preference classifier estimation in OASST but not in HH, as the OASST directly abides by this assumption (outputs are globally ranked) whereas HH does not explicitly. See Appendix L for additional training and evaluation details.

**Implementing WRO-KL.** Variants of WRO-KL can be implemented via different $\psi$ in $\max_\theta \mathbb{E}_{p(\mathbf{x})} \mathbb{E}_{p_\theta(\mathbf{y} \,|\, \mathbf{x})}[\psi(\mathbf{x}, \mathbf{y}) - \beta \log p_\theta(\mathbf{y} \,|\, \mathbf{x}) + \beta \log p_{ref}(\mathbf{y} \,|\, \mathbf{x})]$, e.g., $\psi(\mathbf{x}, \mathbf{y}) = \mathbb{E}_{p(\mathbf{y}_0 \,|\, \mathbf{x})}[p(\ell = 1 \,|\, \mathbf{x}, \mathbf{y}_0, \mathbf{y})]$. We approximate expectations over generator distributions with a single-sample Monte-Carlo estimate. Concretely, for a given query we sample one response each from the current and original model to compute $\psi$, which is either a function of 1. the preference probability under the judge model which takes in the pair as input, or 2. the BT preference probability under the reward model which takes each sequence in as input separately. The one exception is WRO-KL-logit-BT (i.e., RLHF), where we drop $\mathbb{E}_{p(\mathbf{y}_0 \,|\, \mathbf{x})}[r(\mathbf{x}, \mathbf{y}_0)]$ and only optimize $r(\mathbf{x}, \mathbf{y})$ (see Appendix M for a comparison between keeping and dropping this term). For optimization, we use the PPO algorithm from the TRL library (von Werra et al., 2020).

## 6.2. Results: Comparing RLHF, DPO, and SFT

Figure 2 compares methods. While SFT performance aligns with expectations, as does DPO performance with more regularization, RLHF substantially underperforms relative to expected given the aforementioned analysis, as does DPO with less regularization. The non-monotonic nature of both point to the influence of factors beyond target distribution for win rate improvements. To test if RLHF underperformance is due to error from using an estimated reward model instead of the true preference classifier, we additionally run a WRO-logit variant with oracle preference classifier, but this performs even worse, suggesting another factor at play.

Table 1: Win rate of WRO-KL variants over the reference model. Row corresponding to RLHF is shaded grey. No choice of $\hat{p}_\ell$, $h$, or $\beta$ systematically outperforms all others.

| Dataset | $\hat{p}_\ell$ | $h$ | $\beta = 0.001$ | $\beta = 0.01$ | $\beta = 0.1$ | $\beta = 1$ |
|---|---|---|---|---|---|---|
| HH | non-BT | log | 38.36 (1.71) | 46.38 (1.34) | 63.94 (0.94) | 67.00 (0.83) |
| | | logit | 69.15 (0.91) | 41.98 (1.45) | 48.74 (0.97) | 69.15 (0.93) |
| | | identity | 69.94 (0.99) | 69.94 (0.99) | 65.92 (0.95) | 43.21 (0.54) |
| | BT | log | 35.41 (1.98) | 33.80 (1.59) | 41.76 (1.49) | 49.49 (0.60) |
| | | logit | 70.46 (0.87) | 63.93 (1.24) | 69.44 (0.90) | 54.60 (0.67) |
| | | identity | 65.71 (0.95) | 64.59 (0.95) | 69.94 (0.99) | 51.72 (0.59) |
| | oracle | log | 34.98 (1.74) | 52.74 (0.26) | 47.66 (1.09) | 43.29 (0.65) |
| | | logit | 41.99 (1.65) | 55.13 (1.07) | 50.28 (1.21) | 48.55 (0.44) |
| | | identity | 65.34 (0.93) | 66.42 (1.01) | 68.39 (0.94) | 45.14 (0.44) |
| OASST | non-BT | log | 59.51 (3.88) | 59.02 (3.96) | 61.98 (1.40) | 58.63 (1.98) |
| | | logit | 61.98 (3.53) | 54.15 (3.20) | 65.14 (1.20) | 48.69 (1.33) |
| | | identity | 56.65 (3.26) | 54.20 (2.28) | 64.90 (1.28) | 53.64 (2.10) |
| | BT | log | 63.10 (3.48) | 62.80 (3.55) | 60.94 (3.46) | 54.64 (1.26) |
| | | logit | 62.05 (3.65) | 63.95 (3.38) | 66.72 (1.16) | 58.45 (1.82) |
| | | identity | 62.00 (3.55) | 51.98 (2.96) | 54.29 (2.16) | 50.42 (1.38) |
| | oracle | log | 59.32 (3.79) | 60.19 (3.69) | 56.09 (1.65) | 60.09 (1.21) |
| | | logit | 49.55 (3.06) | 44.34 (2.61) | 53.66 (2.87) | 46.15 (1.55) |
| | | identity | 52.13 (2.47) | 66.69 (1.29) | 66.21 (1.22) | 52.31 (1.25) |

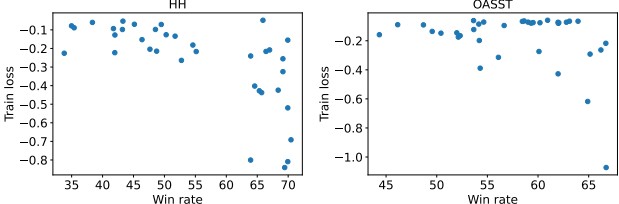

Figure 3: Train loss vs. test win rate across all settings tested in Table 1.

### 6.3. Results: Comparing WRO-KL variants

Table 1 compares different WRO-KL objectives with different choices of $h$, $\beta$, and the estimation of the classifier. Notably, no WRO-KL method (i.e., choice of $h$) outperforms the others systematically across settings. Moreover, the settings that correspond to better target distributions (using a perfect preference classifier, low $\beta$) do not necessarily yield better win rates empirically. These results suggest that there is a more important consideration than the target distribution implied by the objective, namely the success of optimization. Indeed, we see that training loss correlates more with test win rate across WRO-KL than any of the target distribution design choices $\hat{p}_\ell$, $h$, or $\beta$: the p-value of Spearman rank correlation test is 8.27e-5 for train loss vs. win rate, compared to 0.968 and 0.133 for $\hat{p}_\ell$ and $\beta$, and 0.54/0.87/0.87 for all pairs of $h$ choices under Kolmogorov-Smirnov test. Figure 3 plots train loss vs. test win rate for all WRO-KL variants. Interestingly, even though one would expect the losses for different WRO variants to differ in scale under different choices in $h$ and $\beta$, there still exists a surprisingly noticeable global trend of better loss corresponding to better win rate.

## 7. Discussion

Amidst the complex landscape of aligning language models to human preferences, this work offers a simplifying insight:

win rate is the only evaluation that can matter based on preference data alone, and thus all of preference learning should be understood in relation to it (e.g., how well a given method optimizes for win rate, the role of additional assumptions). This work takes the first steps to do that (see Appendix A for summary). High level takeaways from the analysis are that WRO is the theoretical ideal (as existing non-WRO fail win rate-correspondence or consistency), but optimization difficulties can lead to its underperformance in practice. Thus, optimization is a first-order factor in the practical success of WRO, while achieving / improving win rate-correspondence and consistency is important for advancing non-WRO.

**What does this mean for current practice?** The importance and difficulty of optimization for WRO suggests that strategies to improve optimization success could help, e.g., multiple seeds. The lack of win rate-correspondence in DPO/DAA clarifies the importance of checkpointing with metrics other than loss. And SFT's dependence on variance (in preference) suggests that increasing preference diversity of generations to annotate could yield gains.

**What's the value of so many methods?** Interestingly, theory and experiments yield opposing trends: based on win rate-correspondence and consistency, RLHF > DPO > SFT; but based on ease of optimization, SFT > DPO > RLHF.[7] The fact that no existing method is optimal with respect to both the objective being optimized and ease of optimization suggests why it might be useful to combine methods (to take advantage of different strengths) and develop new ones (to strike a better overall balance).

**What's next for preference learning?** Our analysis suggests that the most important improvements in preference learning will likely fall under the umbrella of moving closer to directly optimizing for win rate, either in the objective itself (i.e., coming up with better surrogates for win rate optimization) or the practical optimization of it (e.g., beyond curent policy gradient approaches). How might we make progress? For one, the connection between RLHF / WRO-KL objectives and variational inference suggests the possibility of drawing upon the rich field of probabilistic inference to improve preference learning, from variance reduction techniques (Mohamed et al., 2020b) to alternative optimization objectives and algorithms altogether (Naesseth et al., 2020). Moreover, the strategies and theoretical tools used to understand and develop surrogate objectives for other machine learning tasks (Bridle, 1990; Steinwart, 2007; Awasthi et al., 2022) could provide a useful starting point for developing better objectives for preference learning.

Several concrete directions appear in Section 4 and Section 5 including non-constant sum WRO games to reflect

---

[7]SFT > DPO given failures at low $\beta$ and DPO training issues mentioned in Pal et al. (2024); Chen et al. (2024)

the fact that aligning to huamn preferences need not be a constant-sum endeavor; fully online versions of direct alignment algorithms or contrastive post-training algorithms to help satisfy prevalence consistency (and fewer degrees of freedom for win rate correspondence failures); and alternative filter-plus-SFT style algorithms that take diversity and filter strength into account for solutions with better win rates. There are probably many more promising ideas to explore—as long as win rate remains the guide.

## Impact Statement

This paper presents work whose goal is to advance methods for aligning generative models (especially language models) to human preferences. There are many potential societal consequences of this work; perhaps the most critical is the question of whose preferences are being represented and how.

## Acknowledgements

This work was partly supported by the NIH/NHLBI Award R01HL148248, NSF Award 1922658 NRT-HDR: FUTURE Foundations, Translation, and Responsibility for Data Science, NSF CAREER Award 2145542, ONR N00014-23-1-2634, NSF Award 2404476, Apple, Google, and Meta.

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

## A. Summary of theoretical contributions

Below in Table 2, we present a table summarizing the win rate-centric framework introduced in this work, as well as the original source for each insight within the overall framework:

| | WRO membership | Generalization | Win rate-correspondence | Win rate-consistency |
|---|---|---|---|---|
| RLHF | ✓(Azar et al., 2023) | $h$-win rate (Azar et al., 2023) | ✓(Section 4.3)[†] | ✓(Section 4.3)[†] |
| NLHF | ✓(Munos et al., 2023) | $h$-win rate games (Section 4.2) | ✓(Section 4.3)[†] | ✓(Section 4.3)[†] |
| DPO | ✗(Section 5.1) | direct alignment algorithms (*) | ✗(Section 5.1) | ✓(Rafailov et al., 2024c)[‡] |
| SFT | ✗(Section 5.2) | best-of-n, RAFT, ReST (*) | ✗(Section 5.2) | ✗(Section 5.2) |

Table 2: Summary of the win-rate centric framework introduced in this work. (*) means that the given insight can be attributed to multiple works. Works which generalize DPO and propose alternative direct alignment algorithms: Azar et al. (2023); Huang et al. (2024); Tang et al. (2024); Han et al. (2024). Works that generalize SFT on the preferred sample to more samples include Dong et al. (2023); Gulcehre et al. (2023). [†]RLHF and NLHF typically include a regularization term and satisfy regularized win rate-correspondence and consistency. Without regularization, win rate-correspondence and consistency are satisfied. [‡]DPO only satisfies regularized win rate-consistency ($\beta$ must be greater than zero for a meaningful objective) and was constructed specifically with this goal in mind.

## B. Proposition 3.3 proof

First we show a lemma that breaks down prevalance consistency into a sum over atoms.

**Lemma B.1.** *Let $\beta$ be a functional of a distribution $p$ with support over countable sample space $\mathcal{X}$, where if $p = \alpha p_1 + (1 - \alpha)p_2$, the functional satisfies $\beta(p) = \alpha\beta(p_1) + (1 - \alpha)\beta(p_2)$. Then,*

$$\beta(p) = \sum_{\boldsymbol{x}' \in \mathcal{X}} p(\boldsymbol{x}')\beta(\mathbb{1}[\boldsymbol{x} = \boldsymbol{x}']).$$

*Proof.* We prove this by induction on $n$ the size of the support of $\mathcal{X}$. The base case for $n = 2$ is true by assumption, now assume the statement is true for support size of $n$, then we show it is true for $n + 1$.

Let $\mathcal{X}$ be a support set of size $n + 1$. Let $\mathcal{A}$ be a set containing an element $\mathbf{a}$ in $\mathcal{X}$, where $\alpha = p(\mathbf{x} \in \mathcal{A})$. Then, $p = \alpha p(\mathbf{x} \,|\, \mathbf{x} \in \mathcal{A}) + (1 - \alpha)p(\mathbf{x} \,|\, \mathbf{x} \in \mathcal{A}^c)$ and

$$
\begin{aligned}
\beta(p) &= p(\mathbf{x} \in \mathcal{A})\beta(p(\mathbf{x} \,|\, \mathbf{x} \in \mathcal{A})) + (1 - p(\mathbf{x} \in \mathcal{A}))\beta(p(\mathbf{x} \,|\, \mathbf{x} \in \mathcal{A}^c)) \\
&= p(\mathbf{x} = \mathbf{a})\beta(p(\mathbf{x} \,|\, \mathbf{x} = \mathbf{a}) + (1 - p(\mathbf{x} \in \mathcal{A}))\beta(p(\mathbf{x} \,|\, \mathbf{x} \in \mathcal{A}^c)) \\
&= p(\mathbf{x} = \mathbf{a})\beta(\mathbb{1}[\mathbf{x} = \mathbf{a}]) + (1 - p(\mathbf{x} \in \mathcal{A}))\beta(p(\mathbf{x} \,|\, \mathbf{x} \in \mathcal{A}^c)) \\
&= p(\mathbf{x} = \mathbf{a})\beta(\mathbb{1}[\mathbf{x} = \mathbf{a}]) + (1 - p(\mathbf{x} \in \mathcal{A})) \sum_{\mathbf{x}' \in \mathcal{A}^c} p(\mathbf{x} = \mathbf{x}' \,|\, \mathbf{x} \in \mathcal{A}^c)\beta(\mathbb{1}[\mathbf{x} = \mathbf{x}'])), \quad \mathcal{A}^c \text{ is of size } n \\
&= p(\mathbf{x} = \mathbf{a})\beta(\mathbb{1}[\mathbf{x} = \mathbf{a}]) + \sum_{\mathbf{x}' \in \mathcal{A}^c} p(\mathbf{x} = \mathbf{x}', \mathbf{x} \in \mathcal{A}^c)\beta(\mathbb{1}[\mathbf{x} = \mathbf{x}'])) \\
&= p(\mathbf{x} = \mathbf{a})\beta(\mathbb{1}[\mathbf{x} = \mathbf{a}]) + \sum_{\mathbf{x}' \in \mathcal{A}^c} p(\mathbf{x} = \mathbf{x}')\beta(\mathbb{1}[\mathbf{x} = \mathbf{x}'])) \\
&= \sum_{\mathbf{x}' \in \mathcal{X}} p(\mathbf{x}')\beta(\mathbb{1}[\mathbf{x} = \mathbf{x}']).
\end{aligned}
$$

This completes the induction step. □

**Proposition 3.3.** *For discrete $p$, an evaluation $\phi$ is grounded, as defined in Definition 3.2, if and only if for every generator $p(\boldsymbol{y}_1 \,|\, \boldsymbol{x})$, anchor $p(\boldsymbol{y}_0 \,|\, \boldsymbol{x})$ in query-preference environment $(p(\boldsymbol{x}), p(\ell \,|\, \boldsymbol{x}, \boldsymbol{y}_0, \boldsymbol{y}_1))$*

$$\phi_{p(\boldsymbol{y}_0 \,|\, \boldsymbol{x})}(p(\boldsymbol{y} \,|\, \boldsymbol{x}), \mathcal{E}) = \mathbb{E}_{p(\boldsymbol{x})}\mathbb{E}_{p(\boldsymbol{y} \,|\, \boldsymbol{x})}\mathbb{E}_{p(\boldsymbol{y}_0 \,|\, \boldsymbol{x})}[h \cdot p(\ell = 1 \,|\, \boldsymbol{x}, \boldsymbol{y}_0, \boldsymbol{y})] \tag{8}$$

*Proof.* For the forward direction, we can check both conditions on $\mathbb{E}_{p(\mathbf{x})}\mathbb{E}_{p(\mathbf{y}_1 \,|\, \mathbf{x})}\mathbb{E}_{p(\mathbf{y}_0 \,|\, \mathbf{x})}[h \cdot p(\ell = 1 \,|\, \mathbf{x}, \mathbf{y}_0, \mathbf{y}_1)]$:

- Preference consistency holds because if all distributions are point masses, the expectation becomes an evaluation at those points

- Prevalence consistency holds by the general statement $\mathbb{E}_{\mathbf{z}\sim aq_1+(1-a)q_2}(f(\mathbf{z})) = a\mathbb{E}_{q_1}[f(\mathbf{z})] + (1-a)\mathbb{E}_{q_2}[f(\mathbf{z})]$.

For the reverse, we need to show that if an evaluation $\phi$ meets preference and prevalence consistency, it can be written as $\phi_{p(\mathbf{y}_0\,|\,\mathbf{x})}(p(\mathbf{y}_1\,|\,\mathbf{x}), \mathcal{E}) = \mathbb{E}_{p(\mathbf{x})}\mathbb{E}_{p(\mathbf{y}_1\,|\,\mathbf{x})}\mathbb{E}_{p(\mathbf{y}_0\,|\,\mathbf{x})}[h \cdot p(\ell = 1\,|\,\mathbf{x}, \mathbf{y}_0, \mathbf{y}_1)]$ for some $h$.

By preference consistency for all triples $\mathbf{x}, \mathbf{y}_1, \mathbf{y}_0$ obtained for generator $p(\mathbf{y}_1\,|\,\mathbf{x})$, anchor $p(\mathbf{y}_0\,|\,\mathbf{x})$, and query-preference environment $(p(\mathbf{x}), p(\ell\,|\,\mathbf{x}, \mathbf{y}_0, \mathbf{y}_1))$, the following holds for some $h$:

$$\phi_{\mathbb{1}[\mathbf{y}_0=\mathbf{y}_0']}(\mathbb{1}[\mathbf{y}_1 = \mathbf{y}_1'], (\mathbb{1}[\mathbf{x} = \mathbf{x}'], p_\ell)) = h \cdot p(\ell = 1\,|\,\mathbf{x}', \mathbf{y}_0', \mathbf{y}_1')$$

Then, by prevalence consistency with $\mathbb{X}$ as the support of $p(\mathbf{x})$, $\mathbb{Y}_0^{\mathbf{x}}$ as the support of $p(\mathbf{y}_0\,|\,\mathbf{x})$, and $\mathbb{Y}_1^{\mathbf{x}}$ as the support of $p(\mathbf{y}_1\,|\,\mathbf{x})$,

$$\phi_{p(\mathbf{y}_0\,|\,\mathbf{x})}(p(\mathbf{y}_1\,|\,\mathbf{x}), \mathcal{E}) = \sum_{\mathbf{x}'\in\mathbb{X}} \phi_{p(\mathbf{y}_0\,|\,\mathbf{x})}(p(\mathbf{y}_1\,|\,\mathbf{x}), (\mathbb{1}[\mathbf{x} = \mathbf{x}'], p_\ell))p(\mathbf{x}')$$

$$= \sum_{\mathbf{x}'\in\mathbb{X}}\sum_{\mathbf{y}_0'\in\mathbb{Y}_0^{\mathbf{x}'}} \phi_{\mathbb{1}[\mathbf{y}_0=\mathbf{y}_0']}(p(\mathbf{y}_1\,|\,\mathbf{x}), (\mathbb{1}[\mathbf{x} = \mathbf{x}'], p_\ell))p(\mathbf{y}_0'\,|\,\mathbf{x}')p(\mathbf{x}')$$

$$= \sum_{\mathbf{x}'\in\mathbb{X}}\sum_{\mathbf{y}_0'\in\mathbb{Y}_0^{\mathbf{x}'}}\sum_{\mathbf{y}_1'\in\mathbb{Y}_1^{\mathbf{x}'}} \phi_{\mathbb{1}[\mathbf{y}_0=\mathbf{y}_0']}(\mathbb{1}[\mathbf{y}_1 = \mathbf{y}_1'], (\mathbb{1}[\mathbf{x} = \mathbf{x}'], p_\ell))p(\mathbf{y}_1'\,|\,\mathbf{x}')p(\mathbf{y}_0'\,|\,\mathbf{x}')p(\mathbf{x}')$$

$$= \sum_{\mathbf{x}'\in\mathbb{X}}\sum_{\mathbf{y}_0'\in\mathbb{Y}_0^{\mathbf{x}'}}\sum_{\mathbf{y}_1'\in\mathbb{Y}_1^{\mathbf{x}'}} (h \cdot p(\ell = 1\,|\,\mathbf{x}', \mathbf{y}_0', \mathbf{y}_1'))p(\mathbf{y}_1'\,|\,\mathbf{x}')p(\mathbf{y}_0'\,|\,\mathbf{x}')p(\mathbf{x}'), \quad \text{since preference consistency holds for all atoms}$$

$$= \mathbb{E}_{p(\mathbf{x})}\mathbb{E}_{p(\mathbf{y}\,|\,\mathbf{x})}\mathbb{E}_{p(\mathbf{y}_0\,|\,\mathbf{x})}[h \cdot p(\ell = 1\,|\,\mathbf{x}, \mathbf{y}_0, \mathbf{y})].$$

$\square$

To generalize the results to a continuous density, one would need to additionally assume some notion of convergence for $\phi$ as well to use weak convergence from a discrete measure.

## C. Proposition 4.1 Proof

We provide both the informal statement in the main paper as well as its formal version.

**Proposition 4.1.** (informal) *(informal) Under the Bradley-Terry assumption, all objectives in Equation* (2) *with strictly increasing $h$ share the same optimal solution. This solution maximizes win rate overall all possible anchor distributions.*

**Proposition 4.1.** (formal) *Denote by $\mathcal{P}_h^*$ the set of distributions $p(\mathbf{y}\,|\,\mathbf{x})$ that optimize the WRO-$h$ objective in Equation* (2). *Assume a hypothesis class induced by $\theta \in \Theta$ such that all optima are realizable. Then, for a given anchor distribution $p(\mathbf{y}_0\,|\,\mathbf{x})$, $\mathcal{P}_h^* = \mathcal{P}_{h'}^*$ for any strictly increasing $h$ under the Bradley-Terry assumption with finite rewards.*

*Proof.* We first introduce the $\mathcal{P}_{\text{reward}}^*$, the set of all distributions $p(\mathbf{y}\,|\,\mathbf{x})$ which for each $\mathbf{x}$ place all their probability mass over only the highest-reward sequences or some subset of them. Then, we show that $\mathcal{P}_{\text{reward}}^* = \mathcal{P}_h^*$ for any strictly increasing $h$ and any anchor distribution $p(\mathbf{y}_0\,|\,\mathbf{x})$. In other words, any maximum-reward distribution $p_{\text{reward}}^*(\mathbf{y}\,|\,\mathbf{x}) \in \mathcal{P}_{\text{reward}}^*$ maximizes any WRO objective and vice versa:

$$\mathbb{E}_{p(\mathbf{x})}\mathbb{E}_{p_{\text{reward}}^*(\mathbf{y}\,|\,\mathbf{x})}\mathbb{E}_{p(\mathbf{y}_0\,|\,\mathbf{x})}[h \circ p(\ell = 1\,|\,\mathbf{x}, \mathbf{y}_0, \mathbf{y})] \tag{9}$$

$$= \mathbb{E}_{p(\mathbf{x})}\mathbb{E}_{p_{\text{reward}}^*(\mathbf{y}\,|\,\mathbf{x})}\mathbb{E}_{p(\mathbf{y}_0\,|\,\mathbf{x})}[h \circ \sigma\,(r(\mathbf{x}, \mathbf{y}) - r(\mathbf{x}, \mathbf{y}_0))] \tag{10}$$

$$= \max_{\theta} \mathbb{E}_{p(\mathbf{x})}\mathbb{E}_{p_\theta(\mathbf{y}_1\,|\,\mathbf{x})}\mathbb{E}_{p(\mathbf{y}_0\,|\,\mathbf{x})}[h \circ \sigma\,(r(\mathbf{x}, \mathbf{y}_1) - r(\mathbf{x}, \mathbf{y}_0))] \tag{11}$$

$$= \max_{\theta} \mathbb{E}_{p(\mathbf{x})}\mathbb{E}_{p_\theta(\mathbf{y}_1\,|\,\mathbf{x})}\mathbb{E}_{p(\mathbf{y}_0\,|\,\mathbf{x})}[h \circ p(\ell = 1\,|\,\mathbf{x}, \mathbf{y}_0, \mathbf{y}_1)]. \tag{12}$$

It follows that $\mathcal{P}_h^* = \mathcal{P}_{h'}^*$ for any strictly increasing $h$. $\square$

## D. Target distribution derivations

### D.1. WRO-KL

Here, we show that the WRO-KL objective is equivalent to minimizing the reverse KL divergence of the model and following target distribution:

$$p^*_{\text{WRO-KL}}(\mathbf{y}\,|\,\mathbf{x}) \propto p_{\text{ref}}(\mathbf{y}\,|\,\mathbf{x})\exp(\frac{1}{\beta}\mathbb{E}_{p(\mathbf{y}_0|\mathbf{x})}[h\cdot p(\ell=1|\mathbf{x},\mathbf{y}_0,\mathbf{y})]). \tag{13}$$

Derivation:

$$\max_\theta \mathbb{E}_{p(\mathbf{x})}\big[\mathbb{E}_{p_\theta(\mathbf{y}_1\,|\,\mathbf{x})}\mathbb{E}_{p(\mathbf{y}_0\,|\,\mathbf{x})}[h\cdot p(\ell=1\,|\,\mathbf{x},\mathbf{y}_0,\mathbf{y}_1)] - \beta\text{KL}(p_\theta(\mathbf{y}\,|\,\mathbf{x})\,\|\,p_{\text{ref}}(\mathbf{y}\,|\,\mathbf{x}))\big] \tag{14}$$

$$= \min_\theta -\mathbb{E}_{p(\mathbf{x})}\mathbb{E}_{p_\theta(\mathbf{y}_1\,|\,\mathbf{x})}\left[\mathbb{E}_{p(\mathbf{y}_0\,|\,\mathbf{x})}[h\cdot p(\ell=1\,|\,\mathbf{x},\mathbf{y}_0,\mathbf{y}_1)] - \beta\log\frac{p_\theta(\mathbf{y}\,|\,\mathbf{x})}{p_{\text{ref}}(\mathbf{y}\,|\,\mathbf{x})}\right] \tag{15}$$

$$= \min_\theta \mathbb{E}_{p(\mathbf{x})}\mathbb{E}_{p_\theta(\mathbf{y}_1\,|\,\mathbf{x})}\left[\log\frac{p_\theta(\mathbf{y}\,|\,\mathbf{x})}{p_{\text{ref}}(\mathbf{y}\,|\,\mathbf{x})} - \frac{1}{\beta}\mathbb{E}_{p(\mathbf{y}_0\,|\,\mathbf{x})}[h\cdot p(\ell=1\,|\,\mathbf{x},\mathbf{y}_0,\mathbf{y}_1)]\right] \tag{16}$$

$$= \min_\theta \mathbb{E}_{p(\mathbf{x})}\mathbb{E}_{p_\theta(\mathbf{y}_1\,|\,\mathbf{x})}\left[\log\frac{p_\theta(\mathbf{y}\,|\,\mathbf{x})}{p_{\text{ref}}(\mathbf{y}\,|\,\mathbf{x})\exp(\frac{1}{\beta}\mathbb{E}_{p(\mathbf{y}_0\,|\,\mathbf{x})}[h\cdot p(\ell=1\,|\,\mathbf{x},\mathbf{y}_0,\mathbf{y}_1)])}\right] \tag{17}$$

$$= \min_\theta \mathbb{E}_{p(\mathbf{x})}\mathbb{E}_{p_\theta(\mathbf{y}_1\,|\,\mathbf{x})}\left[\log\frac{p_\theta(\mathbf{y}\,|\,\mathbf{x})}{\frac{1}{Z(\mathbf{x})}p_{\text{ref}}(\mathbf{y}\,|\,\mathbf{x})\exp(\frac{1}{\beta}\mathbb{E}_{p(\mathbf{y}_0\,|\,\mathbf{x})}[h\cdot p(\ell=1\,|\,\mathbf{x},\mathbf{y}_0,\mathbf{y}_1)])} - \log Z(\mathbf{x})\right] \tag{18}$$

$$= \min_\theta \mathbb{E}_{p(\mathbf{x})}\big[\text{KL}(p_\theta(\mathbf{y}\,|\,\mathbf{x})\,\|\,p^*_{\text{WRO-KL}}(\mathbf{y}\,|\,\mathbf{x}))\big], \tag{19}$$

$$p^*_{\text{WRO-KL}}(\mathbf{y}\,|\,\mathbf{x}) \propto p_{\text{ref}}(\mathbf{y}\,|\,\mathbf{x})\exp\left(\frac{1}{\beta}\mathbb{E}_{p(\mathbf{y}_0|\mathbf{x})}[h\cdot p(\ell=1|\mathbf{x},\mathbf{y}_0,\mathbf{y})]\right). \tag{20}$$

### D.2. RLHF

Here, we show that, under the BT assumption, the RLHF objective is equivalent to minimizing the reverse KL divergence with the following target distribution:

$$p^*_{\text{RLHF}}(\mathbf{y}\,|\,\mathbf{x}) \propto p_{\text{ref}}(\mathbf{y}\,|\,\mathbf{x})\exp(\frac{1}{\beta}\mathbb{E}_{p(\mathbf{y}_0|\mathbf{x})}[\text{logit}\, p(\ell=1|\mathbf{x},\mathbf{y}_0,\mathbf{y})]). \tag{21}$$

$$\max_\theta \mathbb{E}_{p(\mathbf{x})}\big[\mathbb{E}_{p_\theta(\mathbf{y}\,|\,\mathbf{x})}[r(\mathbf{x},\mathbf{y})] - \beta\text{KL}(p_\theta(\mathbf{y}\,|\,\mathbf{x})\,\|\,p_{\text{ref}}(\mathbf{y}\,|\,\mathbf{x}))\big] \tag{22}$$

$$= \max_\theta \mathbb{E}_{p(\mathbf{x})}\big[\mathbb{E}_{p_\theta(\mathbf{y}\,|\,\mathbf{x})}\mathbb{E}_{p(\mathbf{y}_0\,|\,\mathbf{x})}[\text{logit}\, p(\ell=1\,|\,\mathbf{x},\mathbf{y}_0,\mathbf{y})] - \beta\text{KL}(p_\theta(\mathbf{y}\,|\,\mathbf{x})\,\|\,p_{\text{ref}}(\mathbf{y}\,|\,\mathbf{x}))\big] \tag{23}$$

$$= \min_\theta \mathbb{E}_{p(\mathbf{x})}\mathbb{E}_{p_\theta(\mathbf{y}\,|\,\mathbf{x})}\left[\log\frac{(p_\theta(\mathbf{y}\,|\,\mathbf{x})}{p_{\text{ref}}(\mathbf{y}\,|\,\mathbf{x}))} - \frac{1}{\beta}\mathbb{E}_{p(\mathbf{y}_0\,|\,\mathbf{x})}[\text{logit}\, p(\ell=1\,|\,\mathbf{x},\mathbf{y}_0,\mathbf{y})]\right] \tag{24}$$

$$= \min_\theta \mathbb{E}_{p(\mathbf{x})}\big[\text{KL}(p_\theta(\mathbf{y}\,|\,\mathbf{x})\,\|\,p^*_{\text{RLHF}}(\mathbf{y}\,|\,\mathbf{x}))\big], \tag{25}$$

$$p^*_{\text{RLHF}}(\mathbf{y}\,|\,\mathbf{x}) \propto p(\mathbf{y}\,|\,\mathbf{x})\exp\left(\frac{1}{\beta}\mathbb{E}_{p(\mathbf{y}_0|\mathbf{x})}[\text{logit}\, p(\ell=1|\mathbf{x},\mathbf{y}_0,\mathbf{y})]\right). \tag{26}$$

## E. Comparing target distributions

In Figure 4, we present a visualization of the target distributions of different preference learning objectives. Panel (a) shows an example preference environment defined by the true preference probabilities between responses (left) and initial starting model (right). Panels (b), (c), and (d) visualize the resulting target distribution of different objectives. In each, the figure on the left shows how the average preference probabilities $\mathbb{E}_{p(\mathbf{y}_0\,|\,\mathbf{x})}[p(\ell=1\,|\,\mathbf{x},\mathbf{y}_0,\mathbf{y}_1]$ (red dots) are translated into the tilt applied to the starting model, i.e. $g(\mathbf{y},\mathbf{x})$ in $p^*(\mathbf{y}\,|\,\mathbf{x}) \propto p(\mathbf{y}\,|\,\mathbf{x})g(\mathbf{y},\mathbf{x})$ (green dots). The figure on the right shows the optimal distribution under the objective. Notably, among the WRO-KL family, the choice of $h$ can make a substantial difference in the target distribution of the objective (panels b vs. c). Moreover, SFT is limited in how much mass it can put on the preferred sample A, as $g(\mathbf{y},\mathbf{x})$ is only the average preference probabilities themselves (panel d).

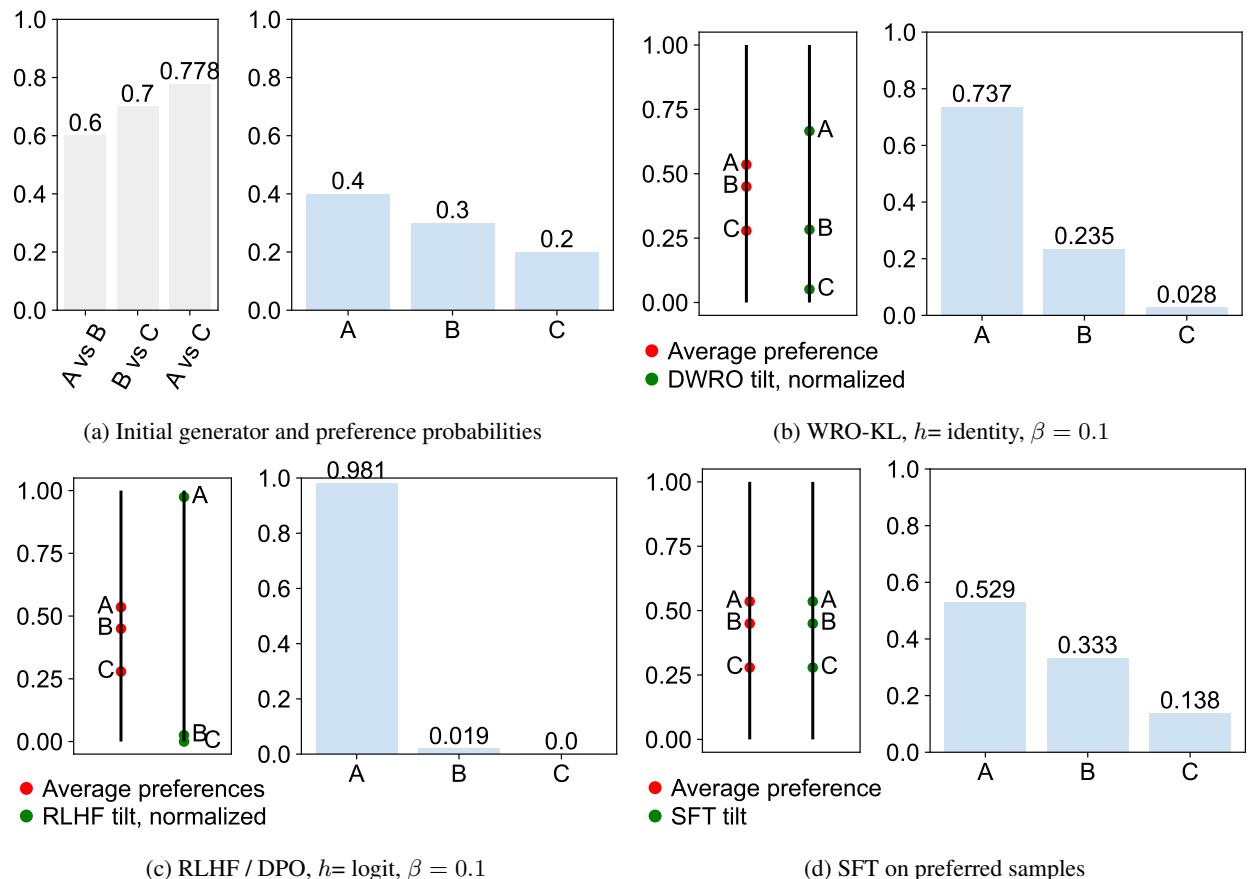

(a) Initial generator and preference probabilities

(b) WRO-KL, $h$= identity, $\beta = 0.1$

(c) RLHF / DPO, $h$= logit, $\beta = 0.1$

(d) SFT on preferred samples

Figure 4: Different preference learning objectives have different target distributions. Consider the initial setting in (a). WRO-KL with $h$= identity, $\beta = 0.1$ yields the target distribution depicted in (b). RLHF ($h$= logit, $\beta = 0.1$) yields a sharper distribution given that average logit probabilities can differ more from each other than average probabilities in (c). SFT yields the least sharp distribution as it can only apply weights between 0 and 1 in (d).

## F. DPO win rate-correspondence counterexamples

### F.1. Example where data is sampled from original model

Consider a query environment $p(\mathbf{x}) = \mathbb{1}[1][\mathbf{x} = \mathbf{x}']$ and a sample space of three responses, $\mathbf{y}_a, \mathbf{y}_b, \mathbf{y}_c$. Under the preference environment of interest, $\mathbf{y}_a$ is preferred over $\mathbf{y}_b$ and both are very preferred over $\mathbf{y}_c$, i.e., $\mathbf{y}_a \succ \mathbf{y}_b \succ\succ \mathbf{y}_c$. Let starting anchor distribution $p_{\text{ref}}(\mathbf{y} \mid \mathbf{x})$ be supported over $\mathbf{y}_a$ and $\mathbf{y}_b$ only, where $p_{\text{ref}}(\mathbf{y}_a \mid \mathbf{x}') > p_{\text{ref}}(\mathbf{y}_b \mid \mathbf{x}')$. The dataset is generated from the initial model itself.

Consider a $p_\theta(\mathbf{y} \mid \mathbf{x})$ that ranks $\mathbf{y}_a$ and $\mathbf{y}_b$ correctly but places most of its mass on $\mathbf{y}_c$. Then, $-\mathcal{L}_{DPO}(p_\theta(\mathbf{y} \mid \mathbf{x}))$ will be high, higher than the $\mathcal{L}_{\text{DPO}}(p_{\text{ref}}(\mathbf{y} \mid \mathbf{x}))$, yet $p_\theta(\mathbf{y} \mid \mathbf{x})$ will generally output a response that is dispreferred over one generated from $p_{\text{ref}}(\mathbf{y} \mid \mathbf{x})$, i.e., $\mathbf{y}_c$ vs. $\mathbf{y}_a$ or $\mathbf{y}_b$. Note that this example even considers online data (i.e., from the initial model) and no estimation error from finite samples; however, even so, the DPO objective does not respect prevalences from the generator itself, which can also be seen as a failing condition 2 Item 2a.

### F.2. Example where data is sampled in a fully online manner

We now consider the DPO objective for each under a fully online setting, i.e., one response sampled from the anchor, another sampled from the generator, and the ability to obtain the oracle preference probability for every response pair. We show below that even then, DPO does not obey win rate-correspondence.

Consider a query environment $p(\mathbf{x}) = \mathbb{1}[1][\mathbf{x} = \mathbf{x}']$ and a sample space of three responses, $\mathbf{y}_a, \mathbf{y}_b, \mathbf{y}_c$, where $\mathbf{y}_a \succ\succ \mathbf{y}_b \succ$

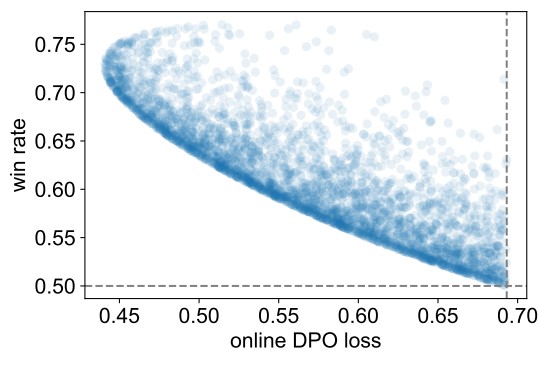
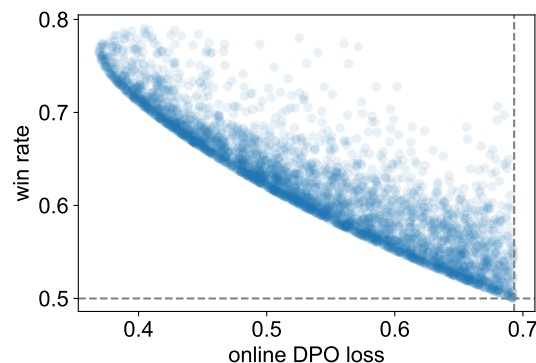

(a) $p(\ell = 1 \mid \mathbf{x}, \mathbf{y}_b, \mathbf{y}_a) = .9$, $p(\ell = 1 \mid \mathbf{x}, \mathbf{y}_c, \mathbf{y}_b) = .6$,
$p(\ell = 1 \mid \mathbf{x}, \mathbf{y}_c, \mathbf{y}_a) \approx .93$.
$p_0(\mathbf{y}_a \mid \mathbf{x}) = p_0(\mathbf{y}_b \mid \mathbf{x}) = p_0(\mathbf{y}_c \mid \mathbf{x}) = 1/3$.

(b) $p(\ell = 1 \mid \mathbf{x}, \mathbf{y}_b, \mathbf{y}_a) = .9$, $p(\ell = 1 \mid \mathbf{x}, \mathbf{y}_c, \mathbf{y}_b) = .9$,
$p(\ell = 1 \mid \mathbf{x}, \mathbf{y}_c, \mathbf{y}_a) \approx .99$.
$p_0(\mathbf{y}_a \mid \mathbf{x}) = p_0(\mathbf{y}_b \mid \mathbf{x}) = p_0(\mathbf{y}_c \mid \mathbf{x}) = 1/3$.

Figure 5: Plot of win rate vs. DPO loss for different $p_\theta(\mathbf{y} \mid \mathbf{x})$ under a given setting of Bradley-Terry preference classifier $p(\ell = 1 \mid \mathbf{x}, \mathbf{y}_0, \mathbf{y}_1)$ and initial and anchor distribution $p_0(\mathbf{y} \mid \mathbf{x})$. These plots are analogous to Figure 1 but with different experimental settings. Each point on the plot is a different instantiation of $p_\theta(\mathbf{y} \mid \mathbf{x})$, sampled from a Dirichlet with $\alpha = 1$ uniformly over all three responses. $\beta = 1$. Grey dotted lines are the values of the win rate (horizontal) and DPO loss (vertical) at initialization. Any two pairs of points that would form a line segment with a positive slope is a pair of generator settings $\theta, \theta'$ such that win rate-correspondence does not hold.

$\mathbf{y}_c$ under the preference environment. Let starting anchor distribution $p_{\text{ref}}(\mathbf{y} \mid \mathbf{x})$ rank the responses as $\mathbf{y}_b \succ \mathbf{y}_c \succ \succ \mathbf{y}_a$, i.e., the rank ordering of $\mathbf{y}_b$ and $\mathbf{y}_c$ is correct, but $\mathbf{y}_a$ is very unlikely relative to them even though it is most preferred. In particular, assume a Bradley-Terry preference classifier that greatly prefers $\mathbf{y}_a$ to $\mathbf{y}_b$, i.e., $p(\ell = 1 \mid \mathbf{x}, \mathbf{y}_b, \mathbf{y}_a) = .9$, and only mildly prefers $\mathbf{y}_b$ to $\mathbf{y}_c$, i.e., $p(\ell = 1 \mid \mathbf{x}, \mathbf{y}_c, \mathbf{y}_b) = .6$. By the BT assumption, $p(\ell = 1 \mid \mathbf{x}, \mathbf{y}_c, \mathbf{y}_a) \approx .93$. Let the original starting model assign probabilities .1, .5, and .4 respectively to $\mathbf{y}_a, \mathbf{y}_b, \mathbf{y}_c$—notably, it's ranking for pairs with $\mathbf{y}_a$ is very wrong but its ranking for $\mathbf{y}_b$ vs. $\mathbf{y}_c$ is correct.

This is the setting used to generate the plot in Figure 1 in the main paper. We generate points on the plot by sampling $p(\theta(\mathbf{y} \mid \mathbf{x}))$ uniformly from the probability simplex. We only plot points whose loss improves upon the initial model (as those are the practical model instances to consider during training), and sample until the plot contains 5000 points.

We now provide a concrete example of $p_\theta(\mathbf{y} \mid \mathbf{x})$ and $q_\theta(\mathbf{y} \mid \mathbf{x})$ where $-\mathcal{L}_{\text{DPO}}(p_\theta) > -\mathcal{L}_{\text{DPO}}(q_\theta)$ yet $\phi(p_\theta) < \phi(q_\theta)$—in words, $p_\theta$ has a better DPO loss but worse win rate than $q_\theta$. Namely, we construct $p_\theta$ such that it continues to increase the correct probability gap between $\mathbf{y}_b$ and $\mathbf{y}_c$ without adding probability mass to $\mathbf{y}_a$, while we construct $q_\theta$ to improve the probability of $\mathbf{y}_a$ but at the expense of a worse probability gap between $\mathbf{y}_b$ and $\mathbf{y}_c$. As a concrete example, set $p_\theta(\mathbf{y}_a \mid \mathbf{x}) = .1$, $p_\theta(\mathbf{y}_b \mid \mathbf{x}) = .6$, $p_\theta(\mathbf{y}_c \mid \mathbf{x}) = .3$, and $q_\theta(\mathbf{y}_a \mid \mathbf{x}) = .8$, $q_\theta(\mathbf{y}_b \mid \mathbf{x}) = .001$, $q_\theta(\mathbf{y}_c \mid \mathbf{x}) = .199$. The online DPO loss for $p_\theta$ and $q_\theta$ is .51 and .78 respectively. The win rate over the original model, however, is .54 and .67 respectively. The logit-win rate (i.e., $h$=logit in Equation (1)) is .26 and 1.7 respectively.

The intuition for large mismatch in this example comes from the differential impact to the DPO loss between improving the ranking of the initially incorrect $\mathbf{y}_a$ vs. $\mathbf{y}_b$ vs. the initially correct $\mathbf{y}_b$ vs. $\mathbf{y}_c$—due to the role of the initial model log probability margins on the loss itself—compared to the differential impact to the win rate between increasing the probability over $\mathbf{y}_a$ versus $\mathbf{y}_b$.

However, this example is just one instance of the mismatch between DPO and win rate; there exist mismatched pairs even when the initial model is uniform over responses, as well as when the BT preference classifier maintains the same probability gap between neighboring examples in the rank order; see Figure 5 for a visualization.

The more fundamental issue is the fact that Equation (4) inside the DPO loss is meant to approximate the preference classifier, but this approximation is only correct when the model is exactly the target distribution for the objective. Then, even in the online setting where the expectations consider the correct distributions, the estimate of the preference classifier is incorrect, i.e., $p_\theta(\ell = 1 \mid \mathbf{x}, \mathbf{y}', \mathbf{y}) \neq p(\ell = 1 \mid \mathbf{x}, \mathbf{y}', \mathbf{y})$, meaning the DPO objective is not the same as win rate under the

preference environment of interest. Namely,

$$-\mathcal{L}_{\text{DPO}}(p_\theta(\mathbf{y}\,|\,\mathbf{x})) = \mathbb{E}_{p(\mathbf{x})p_\theta(\mathbf{y}\,|\,\mathbf{x})p_0(\mathbf{y}'\,|\,\mathbf{x})}[p(\ell=1\,|\,\mathbf{x},\mathbf{y}',\mathbf{y})\log p_\theta(\ell=1\,|\,\mathbf{x},\mathbf{y}',\mathbf{y}) + p(\ell=0\,|\,\mathbf{x},\mathbf{y}',\mathbf{y})\log p_\theta(\ell=0\,|\,\mathbf{x},\mathbf{y}',\mathbf{y})]$$
$$\neq \mathbb{E}_{p(\mathbf{x})p_\theta(\mathbf{y}\,|\,\mathbf{x})p_0(\mathbf{y}'\,|\,\mathbf{x})}[p(\ell=1\,|\,\mathbf{x},\mathbf{y}',\mathbf{y})] = h\text{-Win Rate}_{p(\mathbf{y}_0\,|\,\mathbf{x})}[p_\theta(\mathbf{y}_1\,|\,\mathbf{x})].$$

### F.3. Example that also fails regularized win rate-correspondence

Below is a counterexample to prove that DPO does not obey regularized win rate-correspondence, which is the property that an improvement in the objective implies win rate improvement or a decrease in the divergence to the reference model.

Consider the setting described in the previous section: query distribution is deterministic with a single prompt $\mathbf{x}$, universe of responses to the query are $\{\mathbf{y}_a, \mathbf{y}_b, \mathbf{y}_c\}$, preference classifier follows Bradley-Terry assumption with $p(\ell=1\,|\,\mathbf{x},\mathbf{y}_a,\mathbf{y}_b) = .9$ and $p(\ell=1\,|\,\mathbf{x},\mathbf{y}_b,\mathbf{y}_c) = .6$, and initial model $p_0(\mathbf{y}\,|\,\mathbf{x})$ has probabilities .1, .5, and .4 over $\mathbf{y}_a, \mathbf{y}_b$, and $\mathbf{y}_c$ respectively. Let $p_\theta(\mathbf{y}\,|\,\mathbf{x})$ be and .6, .07, and .33 over $\mathbf{y}_a, \mathbf{y}_b$, and $\mathbf{y}_c$, vs. $q_\theta(\mathbf{y}\,|\,\mathbf{x})$ be .56, .43, and .01 over $\mathbf{y}_a, \mathbf{y}_b$, and $\mathbf{y}_c$. Then, $\mathcal{L}(p_\theta) \approx .59 < \mathcal{L}(q_\theta) \approx .64$, i.e., $p_\theta$ has better DPO loss. However, $p_\theta$ has both worse win rate and larger divergence from the initial model, with win rate and logit-win rate of .69 and 1.2 (vs. .70 and 1.3 for $q_\theta$) and reverse-KL divergence of .87 (vs. .86 for $q_\theta$).

The intuition for this counterexample is that $p_\theta$ has slightly more mass on the best response $\mathbf{y}_a$ than $q_\theta$ does, but has $q_\theta$ has much more mass on the second best $\mathbf{y}_b$ than $p_\theta$, which places the remaining mass on $\mathbf{y}_c$. In other words, $p_\theta$ and $q_\theta$ have different strengths. In this specific setting, $p_\theta$ gets better DPO loss whereas $q_\theta$ gets better win rate over the original model as the small difference in $\mathbf{y}_a$ mass matters less than larger mass difference between $\mathbf{y}_b$ and $\mathbf{y}_c$. Ultimately, $q_\theta$ is slightly closer to the original model, which had most of its mass on $\mathbf{y}_b$.

The broader takeaway is that although DPO has the same target distribution as a regularized WRO objective, the loss is not the same as a linear combination of win rate and explicit regularization term, such that it is possible to improve the DPO objective in a way that does not improve win rate or the regularization.

## G. SFT proofs

We provide results for SFT on the preferred sample in a pair, as well as SFT with arbitrary filtering based on a pair and its preference.

We first provide a more general setting of the theorems in the main paper (Lemma G.1 and Lemma G.2); then we specialize to the setting in the main paper, where $p(\mathbf{y}_0\,|\,\mathbf{x}) = p(\mathbf{y}_1\,|\,\mathbf{x})$, i.e., both samples come from the same distribution.

**Lemma G.1.** *Let $p(\mathbf{y}_0\,|\,\boldsymbol{x})$ be the initial generative model, and $p_{SFT}(\mathbf{y}\,|\,\boldsymbol{x})$ be the target distribution of supervised finetuning on preferred samples $\mathbf{y}_0, \mathbf{y}_1 \sim p(\mathbf{y}_0, \mathbf{y}_1\,|\,\boldsymbol{x}) = p(\mathbf{y}_0\,|\,\boldsymbol{x})p(\mathbf{y}_1\,|\,\boldsymbol{x})$. Then,*

$$\textit{Win Rate}_{p(\mathbf{y}_0\,|\,\boldsymbol{x})}[p_{SFT}(\mathbf{y}\,|\,\boldsymbol{x})] = \tag{27}$$

$$\textit{Win Rate}_{p(\mathbf{y}_0\,|\,\boldsymbol{x})}[p(\mathbf{y}_1\,|\,\boldsymbol{x})] + \int p(\boldsymbol{x})\left[\frac{\textit{Variance}_{p(\mathbf{y}_1\,|\,\boldsymbol{x})}\left[\int p(\mathbf{y}_0\,|\,\boldsymbol{x})p(\ell=1\,|\,\boldsymbol{x},\mathbf{y}_0,\mathbf{y}_1)d\mathbf{y}_0\right]}{\int p(\mathbf{y}_1'\,|\,\boldsymbol{x})\int p(\ell=1|\boldsymbol{x},\mathbf{y}_0'',\mathbf{y}_1')p(\mathbf{y}_0\,|\,\boldsymbol{x})d\mathbf{y}_0''d\mathbf{y}_1'}\right]d\boldsymbol{x}. \tag{28}$$

*Proof.*

$$\text{Win Rate}_{p(\mathbf{y}\,|\,\mathbf{x})}[p_{\text{SFT}}(\mathbf{y}\,|\,\mathbf{x})] = \int p(\mathbf{x})p(\mathbf{y}_0\,|\,\mathbf{x})p_{\text{SFT}}(\mathbf{y}\,|\,\mathbf{x})p(\ell = 1\,|\,\mathbf{x},\mathbf{y}_0,\mathbf{y}_1)\mathbf{y}_0\mathbf{y}_1 d\mathbf{x} \tag{29}$$

$$= \int p(\mathbf{x})p(\mathbf{y}_0\,|\,\mathbf{x})\frac{p(\mathbf{y}_1\,|\,\mathbf{x})\int p(\ell = 1|\mathbf{x},\mathbf{y}_0',\mathbf{y}_1)p(\mathbf{y}_0'\,|\,\mathbf{x})d\mathbf{y}_0'}{\int p(\mathbf{y}_1'\,|\,\mathbf{x})\int p(\ell = 1|\mathbf{x},\mathbf{y}_0'',\mathbf{y}_1')p(\mathbf{y}_0''\,|\,\mathbf{x})d\mathbf{y}_0''d\mathbf{y}_1'}p(\ell = 1\,|\,\mathbf{x},\mathbf{y}_0,\mathbf{y}_1)d\mathbf{y}_0 d\mathbf{y}_1 d\mathbf{x} \tag{30}$$

$$= \int p(\mathbf{x})\left[p(\mathbf{y}_0\,|\,\mathbf{x})\frac{p(\mathbf{y}_1\,|\,\mathbf{x})\int p(\ell = 1|\mathbf{x},\mathbf{y}_0',\mathbf{y}_1)p(\mathbf{y}_0'\,|\,\mathbf{x})d\mathbf{y}_0'}{\int p(\mathbf{y}_1'\,|\,\mathbf{x})\int p(\ell = 1|\mathbf{x},\mathbf{y}_0'',\mathbf{y}_1')p(\mathbf{y}_0''\,|\,\mathbf{x})d\mathbf{y}_0''d\mathbf{y}_1'}p(\ell = 1\,|\,\mathbf{x},\mathbf{y}_0,\mathbf{y}_1)d\mathbf{y}_0 d\mathbf{y}_1\right] d\mathbf{x} \tag{31}$$

$$= \int p(\mathbf{x})\left[\int p(\mathbf{y}_0\,|\,\mathbf{x})\frac{p(\mathbf{y}_1\,|\,\mathbf{x})\int p(\ell = 1|\mathbf{x},\mathbf{y}_0',\mathbf{y}_1)p(\mathbf{y}_0'\,|\,\mathbf{x})d\mathbf{y}_0'}{\int p(\mathbf{y}_1'\,|\,\mathbf{x})\int p(\ell = 1|\mathbf{x},\mathbf{y}_0'',\mathbf{y}_1')p(\mathbf{y}_0''\,|\,\mathbf{x})d\mathbf{y}_0''d\mathbf{y}_1'}p(\ell = 1\,|\,\mathbf{x},\mathbf{y}_0,\mathbf{y}_1)d\mathbf{y}_0 d\mathbf{y}_1\right] d\mathbf{x} \tag{32}$$

$$= \int p(\mathbf{x})\left[\int \frac{\int p(\mathbf{y}_0\,|\,\mathbf{x})p(\ell = 1\,|\,\mathbf{x},\mathbf{y}_0,\mathbf{y}_1)d\mathbf{y}_0 p(\mathbf{y}_1\,|\,\mathbf{x})\int p(\ell = 1|\mathbf{x},\mathbf{y}_0',\mathbf{y}_1)p(\mathbf{y}_0'\,|\,\mathbf{x})d\mathbf{y}_0'}{\int p(\mathbf{y}_1'\,|\,\mathbf{x})\int p(\ell = 1|\mathbf{x},\mathbf{y}_0'',\mathbf{y}_1')p(\mathbf{y}_0\,|\,\mathbf{x})d\mathbf{y}_0''d\mathbf{y}_1'}d\mathbf{y}_1\right] d\mathbf{x} \tag{33}$$

$$= \int p(\mathbf{x})\left[\frac{\int (\int p(\mathbf{y}_0\,|\,\mathbf{x})p(\ell = 1\,|\,\mathbf{x},\mathbf{y}_0,\mathbf{y}_1)d\mathbf{y}_0)^2 p(\mathbf{y}_1\,|\,\mathbf{x})d\mathbf{y}_1}{\int p(\mathbf{y}_1'\,|\,\mathbf{x})\int p(\ell = 1|\mathbf{x},\mathbf{y}_0'',\mathbf{y}_1')p(\mathbf{y}_0\,|\,\mathbf{x})d\mathbf{y}_0''d\mathbf{y}_1'}\right] d\mathbf{x} \tag{34}$$

$$= \int p(\mathbf{x})\left[\frac{(\int (\int p(\mathbf{y}_0\,|\,\mathbf{x})p(\ell = 1\,|\,\mathbf{x},\mathbf{y}_0,\mathbf{y}_1)d\mathbf{y}_0)p(\mathbf{y}_1\,|\,\mathbf{x})d\mathbf{y}_1)^2}{\int p(\mathbf{y}_1'\,|\,\mathbf{x})\int p(\ell = 1|\mathbf{x},\mathbf{y}_0'',\mathbf{y}_1')p(\mathbf{y}_0\,|\,\mathbf{x})d\mathbf{y}_0''d\mathbf{y}_1'}\right] d\mathbf{x} \tag{35}$$

$$+ \int p(\mathbf{x})\left[\frac{\text{Variance}_{p(\mathbf{y}_1\,|\,\mathbf{x})}\left[\int p(\mathbf{y}_0\,|\,\mathbf{x})p(\ell = 1\,|\,\mathbf{x},\mathbf{y}_0,\mathbf{y}_1)d\mathbf{y}_0)\right]}{\int p(\mathbf{y}_1'\,|\,\mathbf{x})\int p(\ell = 1|\mathbf{x},\mathbf{y}_0'',\mathbf{y}_1')p(\mathbf{y}_0\,|\,\mathbf{x})d\mathbf{y}_0''d\mathbf{y}_1'}\right] d\mathbf{x} \tag{36}$$

$$= \int p(\mathbf{x})(\int p(\mathbf{y}_0\,|\,\mathbf{x})p(\ell = 1\,|\,\mathbf{x},\mathbf{y}_0,\mathbf{y}_1)d\mathbf{y}_0)p(\mathbf{y}_1\,|\,\mathbf{x})d\mathbf{y}_1 d\mathbf{x} \tag{37}$$

$$+ \int p(\mathbf{x})\left[\frac{\text{Variance}_{p(\mathbf{y}_1\,|\,\mathbf{x})}\left[\int p(\mathbf{y}_0\,|\,\mathbf{x})p(\ell = 1\,|\,\mathbf{x},\mathbf{y}_0,\mathbf{y}_1)d\mathbf{y}_0)\right]}{\int p(\mathbf{y}_1'\,|\,\mathbf{x})\int p(\ell = 1|\mathbf{x},\mathbf{y}_0'',\mathbf{y}_1')p(\mathbf{y}_0\,|\,\mathbf{x})d\mathbf{y}_0''d\mathbf{y}_1'}\right] d\mathbf{x} \tag{38}$$

$$= \text{Win Rate}_{p(\mathbf{y}_0\,|\,\mathbf{x})}[p(\mathbf{y}_1\,|\,\mathbf{x})] + \int p(\mathbf{x})\left[\frac{\text{Variance}_{p(\mathbf{y}_1\,|\,\mathbf{x})}\left[\int p(\mathbf{y}_0\,|\,\mathbf{x})p(\ell = 1\,|\,\mathbf{x},\mathbf{y}_0,\mathbf{y}_1)d\mathbf{y}_0)\right]}{\int p(\mathbf{y}_1'\,|\,\mathbf{x})\int p(\ell = 1|\mathbf{x},\mathbf{y}_0'',\mathbf{y}_1')p(\mathbf{y}_0\,|\,\mathbf{x})d\mathbf{y}_0''d\mathbf{y}_1'}\right] d\mathbf{x}. \tag{39}$$

$\square$

**Lemma G.2.** *Let $p(\mathbf{y}_0\,|\,\boldsymbol{x})$ be the initial generative model, and $p_{\text{filter}}(\mathbf{y}\,|\,\boldsymbol{x})$ be the target distribution of supervised finetuning on samples where the filter $f = 1$ is a potentially randomized data point filter $p(f\,|\,\boldsymbol{x},\mathbf{y}_1,\mathbf{y}_0,\ell)$ that selects data for supervised finetuning based on the any of the context observed $\boldsymbol{x}$, generations $\mathbf{y}_1,\mathbf{y}_0$ and preference $\ell$.*

$$p_{\text{filter}}(\mathbf{y}\,|\,\boldsymbol{x}) = p(\mathbf{y}_1\,|\,\boldsymbol{x}, f = 1) = \frac{p(\mathbf{y}_1\,|\,\boldsymbol{x})p(f = 1\,|\,\boldsymbol{x},\mathbf{y}_1)}{p(f = 1\,|\,\boldsymbol{x})} = \frac{p(\mathbf{y}_1\,|\,\boldsymbol{x})\mathbb{E}_{p(\ell,\mathbf{y}_0\,|\,\boldsymbol{x},\mathbf{y}_1)}[p(f = 1\,|\,\boldsymbol{x},\mathbf{y}_1,\mathbf{y}_0,\ell)]}{\mathbb{E}_{p(\mathbf{y}_1,\mathbf{y}_0,\ell\,|\,\boldsymbol{x})}[p(f = 1\,|\,\boldsymbol{x},\mathbf{y}_1,\mathbf{y}_0,\ell)]}$$

*Then,*

$$\text{Win Rate}_{p(\mathbf{y}_0\,|\,\boldsymbol{x})}[p_{\text{filter}}(\mathbf{y}\,|\,\boldsymbol{x})] =$$

$$\text{Win Rate}_{p(\mathbf{y}_0\,|\,\boldsymbol{x})}[p(\mathbf{y}_1\,|\,\boldsymbol{x})] + \int p(\boldsymbol{x})\left[\frac{Cov_{p(\mathbf{y}_1\,|\,\boldsymbol{x})}(\mathbb{E}_{p(\ell,\mathbf{y}_0\,|\,\boldsymbol{x})}[p(f = 1\,|\,\boldsymbol{x},\mathbf{y}_0,\mathbf{y}_1)],\mathbb{E}_{p(\mathbf{y}_0'\,|\,\boldsymbol{x})}[p(\ell = 1|\boldsymbol{x},\mathbf{y}_0',\mathbf{y}_1)])}{\mathbb{E}_{p(\mathbf{y}_1,\mathbf{y}_0,\ell\,|\,\boldsymbol{x})}[p(f = 1\,|\,\boldsymbol{x},\mathbf{y}_1,\mathbf{y}_0,\ell)]}\right] d\boldsymbol{x}.$$

*Proof.*

$$\text{Win Rate}_{p(\mathbf{y}\,|\,\mathbf{x})}[p_{\text{filter}}(\mathbf{y}\,|\,\mathbf{x})] = \int p(\mathbf{x})p(\mathbf{y}_0\,|\,\mathbf{x})p_{\text{filter}}(\mathbf{y}\,|\,\mathbf{x})p(\ell = 1\,|\,\mathbf{x},\mathbf{y}_0,\mathbf{y}_1)\mathbf{y}_0\mathbf{y}_1 d\mathbf{x}$$

$$= \int p(\mathbf{x})p(\mathbf{y}_0\,|\,\mathbf{x})\frac{p(\mathbf{y}_1\,|\,\mathbf{x})\mathbb{E}_{p(\ell,\mathbf{y}_0\,|\,\mathbf{x},\mathbf{y}_1)}[p(f = 1\,|\,\mathbf{x},\mathbf{y}_1,\mathbf{y}_0,\ell)]}{\mathbb{E}_{p(\mathbf{y}_1,\mathbf{y}_0,\ell\,|\,\mathbf{x})}[p(f = 1\,|\,\mathbf{x},\mathbf{y}_1,\mathbf{y}_0,\ell)]}p(\ell = 1\,|\,\mathbf{x},\mathbf{y}_0,\mathbf{y}_1)d\mathbf{y}_0 d\mathbf{y}_1 d\mathbf{x}$$

$$= \int p(\mathbf{x})\left[\int p(\mathbf{y}_0\,|\,\mathbf{x})\frac{p(\mathbf{y}_1\,|\,\mathbf{x})\mathbb{E}_{p(\ell,\mathbf{y}_0\,|\,\mathbf{x},\mathbf{y}_1)}[p(f = 1\,|\,\mathbf{x},\mathbf{y}_1,\mathbf{y}_0,\ell)]}{\mathbb{E}_{p(\mathbf{y}_1,\mathbf{y}_0,\ell\,|\,\mathbf{x})}[p(f = 1\,|\,\mathbf{x},\mathbf{y}_1,\mathbf{y}_0,\ell)]}p(\ell = 1\,|\,\mathbf{x},\mathbf{y}_0,\mathbf{y}_1)d\mathbf{y}_0 d\mathbf{y}_1\right]d\mathbf{x}$$

$$= \int p(\mathbf{x})\left[\int \frac{p(\mathbf{y}_1\,|\,\mathbf{x})\mathbb{E}_{p(\ell,\mathbf{y}_0\,|\,\mathbf{x},\mathbf{y}_1)}[p(f = 1\,|\,\mathbf{x},\mathbf{y}_1,\mathbf{y}_0,\ell)]\mathbb{E}_{p(\mathbf{y}_0\,|\,\mathbf{x})}[p(\ell = 1\,|\,\mathbf{x},\mathbf{y}_0,\mathbf{y}_1)]}{\mathbb{E}_{p(\mathbf{y}_1,\mathbf{y}_0,\ell\,|\,\mathbf{x})}[p(f = 1\,|\,\mathbf{x},\mathbf{y}_1,\mathbf{y}_0,\ell)]}d\mathbf{y}_1\right]d\mathbf{x}$$

$$= \int p(\mathbf{x})\left[\frac{\mathbb{E}_{p(\mathbf{y}_1\,|\,\mathbf{x})}\left[\mathbb{E}_{p(\ell,\mathbf{y}_0\,|\,\mathbf{x},\mathbf{y}_1)}[p(f = 1\,|\,\mathbf{x},\mathbf{y}_1,\mathbf{y}_0,\ell)]\mathbb{E}_{p(\mathbf{y}_0'\,|\,\mathbf{x})}[p(\ell = 1|\mathbf{x},\mathbf{y}_0',\mathbf{y}_1)]\right]}{\mathbb{E}_{p(\mathbf{y}_1,\mathbf{y}_0,\ell\,|\,\mathbf{x})}[p(f = 1\,|\,\mathbf{x},\mathbf{y}_1,\mathbf{y}_0,\ell)]}d\mathbf{y}_1\right]d\mathbf{x}$$

$$= \int p(\mathbf{x})\left[\frac{\text{Cov}_{p(\mathbf{y}_1\,|\,\mathbf{x})}(\mathbb{E}_{p(\ell,\mathbf{y}_0\,|\,\mathbf{x},\mathbf{y}_1)}[p(f = 1\,|\,\mathbf{x},\mathbf{y}_1,\mathbf{y}_0,\ell)],\mathbb{E}_{p(\mathbf{y}_0'\,|\,\mathbf{x})}[p(\ell = 1|\mathbf{x},\mathbf{y}_0',\mathbf{y}_1)])}{\mathbb{E}_{p(\mathbf{y}_1,\mathbf{y}_0,\ell\,|\,\mathbf{x})}[p(f = 1\,|\,\mathbf{x},\mathbf{y}_1,\mathbf{y}_0,\ell)]}\right]d\mathbf{x}$$

$$+ \int p(\mathbf{x})\left[\frac{\mathbb{E}_{p(\ell,\mathbf{y}_0,\mathbf{y}_1\,|\,\mathbf{x})}[p(f = 1\,|\,\mathbf{x},\mathbf{y}_1,\mathbf{y}_0,\ell)]\mathbb{E}_{p(\mathbf{y}_1\mathbf{y}_0\,|\,\mathbf{x})}[p(\ell = 1|\mathbf{x},\mathbf{y}_0,\mathbf{y}_1)]}{\mathbb{E}_{p(\mathbf{y}_1,\mathbf{y}_0,\ell\,|\,\mathbf{x})}[p(f = 1\,|\,\mathbf{x},\mathbf{y}_1,\mathbf{y}_0,\ell)]}\right]d\mathbf{x}$$

$$= \int p(\mathbf{x})\left[\frac{\text{Cov}_{p(\mathbf{y}_1\,|\,\mathbf{x})}(\mathbb{E}_{p(\ell,\mathbf{y}_0\,|\,\mathbf{x})}[p(f = 1\,|\,\mathbf{x},\mathbf{y}_0,\mathbf{y}_1)],\mathbb{E}_{p(\mathbf{y}_0'\,|\,\mathbf{x})}[p(\ell = 1|\mathbf{x},\mathbf{y}_0',\mathbf{y}_1)])}{\mathbb{E}_{p(\mathbf{y}_1,\mathbf{y}_0,\ell\,|\,\mathbf{x})}[p(f = 1\,|\,\mathbf{x},\mathbf{y}_1,\mathbf{y}_0,\ell)]}\right]d\mathbf{x} + \text{Win Rate}_{p(\mathbf{y}_0\,|\,\mathbf{x})}[p(\mathbf{y}_1\,|\,\mathbf{x})]$$

$\square$

For SFT on the preferred sample, $p(f = 1\,|\,\mathbf{x},\mathbf{y}_1,\mathbf{y}_0,\ell) = \mathbf{1}[\ell = 1]$. Other options which decrease the denominator in the second term while maintaining covariance (e.g., quantile filtering based on $p(\ell = 1\,|\,\mathbf{x},\mathbf{y},\mathbf{y}_1)$) have the potential to enable further win rate improvements.

**Theorem 5.1.** (Win rate improvement of SFT on preferred sample) *Let $p(\mathbf{y}_0\,|\,\mathbf{x})$ be the initial generative model, and $p_{SFT}(\mathbf{y}\,|\,\mathbf{x})$ be the target distribution of supervised finetuning on preferred samples ($p(\mathbf{y}_1\,|\,\mathbf{x},\ell = 1)$, $p(\mathbf{y}_0\,|\,\mathbf{x}) = p(\mathbf{y}_1\,|\,\mathbf{x})$). Then,*

$$\text{Win Rate}_{p(\mathbf{y}_0\,|\,\mathbf{x})}[p_{SFT}(\mathbf{y}\,|\,\mathbf{x})] = 0.5 + 2\mathbb{E}_{p(\mathbf{x})}Var_{p(\mathbf{y}_1\,|\,\mathbf{x})}\left[\int p(\mathbf{y}_0\,|\,\mathbf{x})p(\ell = 1\,|\,\mathbf{x},\mathbf{y}_0,\mathbf{y}_1)d\mathbf{y}_0\right], \tag{40}$$

*which is less than 1.0 as long as there exist at least three responses in $supp(p(\mathbf{y}\,|\,\mathbf{x}))$ for any $\mathbf{x}$.*

*Proof.* We first use the result of Lemma G.1 and plug in the condition $p(\mathbf{y}_0\,|\,\mathbf{x}) = p(\mathbf{y}_1\,|\,\mathbf{x})$:

$$\text{Win Rate}_{p(\mathbf{y}_0\,|\,\mathbf{x})}[p_{SFT}(\mathbf{y}\,|\,\mathbf{x})] \tag{41}$$

$$= 0.5 + \int p(\mathbf{x})\left[\frac{\text{Variance}_{p(\mathbf{y}_1\,|\,\mathbf{x})}\left[\int p(\mathbf{y}_0\,|\,\mathbf{x})p(\ell = 1\,|\,\mathbf{x},\mathbf{y}_0,\mathbf{y}_1)d\mathbf{y}_0)\right]}{\int p(\mathbf{y}_1'\,|\,\mathbf{x})\int p(\ell = 1|\mathbf{x},\mathbf{y}_0'',\mathbf{y}_1')p(\mathbf{y}_0\,|\,\mathbf{x})d\mathbf{y}_0''d\mathbf{y}_1'}\right]d\mathbf{x} \tag{42}$$

$$= 0.5 + 2\int p(\mathbf{x})\text{Variance}_{p(\mathbf{y}_1\,|\,\mathbf{x})}\left[\int p(\mathbf{y}_0\,|\,\mathbf{x})p(\ell = 1\,|\,\mathbf{x},\mathbf{y}_0,\mathbf{y}_1)d\mathbf{y}_0)\right]d\mathbf{x}. \tag{43}$$

$\int p(\mathbf{y}_0\,|\,\mathbf{x})p(\ell = 1\,|\,\mathbf{x},\mathbf{y}_0,\mathbf{y}_1)d\mathbf{y}_0$ can only take values between 0 and 1. Win rate is maximized for a random variable taking values between zero and one when the variance is .25, but Proposition H.1 proves that this is unachievable as long as $\int p(\mathbf{y}_0\,|\,\mathbf{x})p(\ell = 1\,|\,\mathbf{x},\mathbf{y}_0,\mathbf{y}_1)d\mathbf{y}_0 \in (0,1)$ for any $\mathbf{y}_1$. Proposition H.2 shows that $\int p(\mathbf{y}_0\,|\,\mathbf{x})p(\ell = 1\,|\,\mathbf{x},\mathbf{y}_0,\mathbf{y}_1)d\mathbf{y}_0 \in (0,1)$ as long as there exist three or more elements in $supp(p(\mathbf{y}\,|\,\mathbf{x}))$. Thus, as long as there exist at least three responses in $supp(p(\mathbf{y}\,|\,\mathbf{x}))$ for any $\mathbf{x}$, win rate ove the initial model is less than 1. $\square$

Note that under deterministic and transitive preferences that are evenly spaced apart, Theorem 5.1 matches the win rate result in Theorem 2 reported in (Gui et al., 2024) for best-of-n when $n = 2$. In other words, Theorem 5.1 generalizes the best-of-n result to general preference probabilities.

**Theorem 5.2.** (Win rate improvement of filter + SFT) Let $p(f \mid \mathbf{x}, \mathbf{y}_1, \mathbf{y}_0, \ell)$ be a filter that selects data ($f = 1$) for supervised finetuning, and $p(\mathbf{y}_0 \mid \mathbf{x}) = p(\mathbf{y}_1 \mid \mathbf{x})$). Then, denoting $\mathbb{E}_{p(\ell, \mathbf{y}_0 \mid \mathbf{x})} [p(f = 1 \mid \mathbf{x}, \mathbf{y}_0, \mathbf{y}_1, \ell)]$ as $\mathrm{AvgFilter}(\mathbf{x}, \mathbf{y}_1)$ and $\mathbb{E}_{p(\mathbf{y}_0 \mid \mathbf{x})} [p(\ell = 1 \mid \mathbf{x}, \mathbf{y}_0, \mathbf{y}_1)]$ as $\mathrm{AvgPref}(\mathbf{x}, \mathbf{y}_1)$,

$$\text{Win Rate}_{p(\mathbf{y}_0 \mid \mathbf{x})}[p_{\mathrm{SFT}}(\mathbf{y} \mid \mathbf{x})] = 0.5 + \mathbb{E}_{p(\mathbf{x})} \frac{\mathrm{Cov}_{p(\mathbf{y}_1 \mid \mathbf{x})}[\mathrm{AvgFilter}(\mathbf{x}, \mathbf{y}_1), \mathrm{AvgPref}(\mathbf{x}, \mathbf{y}_1)]}{\mathbb{E}_{p(\mathbf{y}_1 \mid \mathbf{x})}\mathrm{AvgFilter}(\mathbf{x}, \mathbf{y}_1)}. \tag{44}$$

*Proof.* We use the result of Lemma G.2 with the condition $p(\mathbf{y}_0 \mid \mathbf{x}) = p(\mathbf{y}_1 \mid \mathbf{x})$. $\qquad \square$

# H. Additional SFT Propositions

Below, Proposition H.1 and Proposition H.2 support Theorem 5.1, the result that the win rate improvement for SFT on preferred samples is limited.

**Proposition H.1.** *Let $X$ be a random variable over [0, 1]. Then, $\mathrm{Var}(X) < .25$ if $\mathrm{Pr}(0 < X < 1) > 0$.*

*Proof.* First, recall that for convex function $f$, $f(x) = f((1 - x) \cdot 0 + x \cdot 1) \le (1 - x) \cdot f(0) + x \cdot f(1)$. Then, letting $f(x) = (x - \mu)^2$, $\mu = \mathbb{E}X$, the variance of a random variable is upper bounded as follows:

$$\begin{aligned}
\mathrm{Var}(X) &= \mathbb{E}[(X - \mu)^2] \\
&\le \mathbb{E}[(1 - X) \cdot f(0) + X \cdot f(1)] \\
&= \mathbb{E}[(1 - X)\mu^2 + X(1 - \mu)^2] \\
&= (1 - \mu)\mu^2 + \mu(1 - \mu)^2 \\
&= \mu^2 - \mu^3 + \mu - 2\mu^2 + \mu^3 \\
&= \mu - \mu^2 = \mu(1 - \mu).
\end{aligned}$$

For a distribution over [0, 1], this quantity is maximized when $\mu = .5$, resulting in a variance of .25. This variance is achieved when $X$ places mass on only 0 and 1 symmetrically: $\mathrm{Var}(X) = .5(0 - .5)^2 + .5(1 - .5)^2 = .25$. Any change to this distribution that changes $\mu$ will have a lower upper bound and thus not achieve variance .25, and any other change that does not change $\mu$ but moves probability mass away from the endpoints will only decrease variance. $\qquad \square$

Since the win-rate is $\text{Win Rate}_{p(\mathbf{y}_0 \mid \mathbf{x})}[p_{\mathrm{SFT}}(\mathbf{y} \mid \mathbf{x})] = 0.5 + 2\mathbb{E}_{p(\mathbf{x})}\mathrm{Var}_{p(\mathbf{y}_1 \mid \mathbf{x})} \left[\mathbb{E}_{p(\mathbf{y}_0 \mid \mathbf{x})} [p(\ell = 1 \mid \mathbf{x}, \mathbf{y}_0, \mathbf{y}_1)]\right]$, the former proposition establishes that $\mathbb{E}_{p(\mathbf{y}_0 \mid \mathbf{x})} [p(\ell = 1 \mid \mathbf{x}, \mathbf{y}_0, \mathbf{y}_1)]$ has to be zero or one in order for win rate to be 1. For a fixed $\mathbf{y}_1$, this expectation being one implies

$$\mathbb{E}_{p(\mathbf{y}_0 \mid \mathbf{x})} [p(\ell = 1 \mid \mathbf{x}, \mathbf{y}_0, \mathbf{y}_1)] = 1 \to p(\ell = 1 \mid \mathbf{x}, \mathbf{y}_0, \mathbf{y}_1) = 1, \{\forall \mathbf{y}_0 \in A\}, \text{for an } A \text{ such that } \mathbb{E}_{p(\mathbf{y}_0 \mid \mathbf{x})}[\mathbb{1}[\mathbf{y}_0 \in A]] = 1$$

More plainly, $\mathbf{y}_1$ must always be preferred over every generation with probability one. By symmetry, the expectation being zero implies that $\mathbf{y}_1$ must always be dispreferred with probability one over every generation.

Being always dispreferred or preferred is impossible due to self-comparison, e.g., for a fixed $\mathbf{y}_1$, the expectation over $p(\mathbf{y}_0 \mid \mathbf{x})$ will include $\mathbf{y}_1$ in the support when samples are generated from the same model. Omitting self-comparisons, as long as the model has support over more than two generations, $\mathbb{E}_{p(\mathbf{y}_0 \mid \mathbf{x})} [p(\ell = 1 \mid \mathbf{x}, \mathbf{y}_0, \mathbf{y}_1)]$ must be less than one for at least one $\mathbf{y}_1 \in \mathrm{supp}(p(\mathbf{y}_1 \mid \mathbf{x}))$.

**Proposition H.2.** *Let $p(\mathbf{y} \mid \mathbf{x})$ have support on three or more elements. Then there exists a $\beta$ in the support of $p(\mathbf{y} \mid \mathbf{x})$ that is neither always preferred or dispreferred over every element in the support of $p(\mathbf{y} \mid \mathbf{x})$.*

*Proof.* Let $\alpha$ be an element that is deterministically preferred over every element in the support of $p(\mathbf{y} \mid \mathbf{x})$, then by symmetry of $(\ell = 1 \mid \mathbf{x}, \mathbf{y}_0, \mathbf{y}_1)$ no other element can be deterministically preferred over every other element. Preferences for $\alpha$: $p(\ell = 1 \mid \mathbf{x}, \mathbf{y}_1 = \alpha, \mathbf{y}_0 = \beta) = 1$ and $p(\ell = 1 \mid \mathbf{x}, \mathbf{y}_1 = \alpha, \mathbf{y}_0 = \gamma) = 1$.

Then let $\gamma$ be an element that is deterministically dispreferred over every element in the support of $p(\mathbf{y} \mid \mathbf{x})$. Preferences for $\gamma$: $p(\ell = 1 \mid \mathbf{x}, \mathbf{y}_1 = \gamma, \mathbf{y}_0 = \alpha) = 0$ and $p(\ell = 1 \mid \mathbf{x}, \mathbf{y}_1 = \gamma, \mathbf{y}_0 = \beta) = 0$. The preference probabilities for $\beta$ are fixed by symmetry from the above assumptions

- Since $p(\ell = 1 \,|\, \mathbf{x}, \mathbf{y}_1 = \alpha, \mathbf{y}_0 = \beta) = 1$, then $p(\ell = 1 \,|\, \mathbf{x}, \mathbf{y}_1 = \alpha, \mathbf{y}_1 = \gamma) = 0$

- Since $p(\ell = 1 \,|\, \mathbf{x}, \mathbf{y}_1 = \gamma, \mathbf{y}_0 = \beta) = 0$, $p(\ell = 1 \,|\, \mathbf{x}, \mathbf{y}_1 = \beta, \mathbf{y}_0 = \gamma) = 1$

Thus, the $\beta$ generation beta is not deterministically preferred or dispreferred. $\qquad\square$

Since being always determinstically preferred or dispreferred is generally not possible, SFT's improvement is limited.

## I. Expected Win Rate Improvement Expressions

Below, we present the expected win rate improvement over the original model for WRO-KL objectives. Letting $p(\mathbf{y}_0 \,|\, \mathbf{x}) = p(\mathbf{y}_0' \,|\, \mathbf{x}) = p(\mathbf{y}_0'' \,|\, \mathbf{x})$ and $p(\mathbf{y}_1 \,|\, \mathbf{x}) = p(\mathbf{y}_1' \,|\, \mathbf{x})$, we have:

$$\text{Win Rate}_{p(\mathbf{y}_0 \,|\, \mathbf{x})}[p_{\text{WRO-KL}}(\mathbf{y} \,|\, \mathbf{x})]$$
$$= \int p(\mathbf{x}) \left[ \frac{\mathbb{E}_{p(\mathbf{y}_1 \,|\, \mathbf{x})} \left[ \mathbb{E}_{p(\mathbf{y}_0 \,|\, \mathbf{x})} p(\ell = 1 \,|\, \mathbf{x}, \mathbf{y}_0, \mathbf{y}_1) \exp\left( \frac{1}{\beta} \mathbb{E}_{p(\mathbf{y}_0' \,|\, \mathbf{x})} h \cdot p(\ell = 1 \,|\, \mathbf{x}, \mathbf{y}_0', \mathbf{y}_1) \right) \right]}{\mathbb{E}_{p(\mathbf{y}_1' \,|\, \mathbf{x})} \exp\left( \frac{1}{\beta} \mathbb{E}_{p(\mathbf{y}_0'' \,|\, \mathbf{x})} h \cdot p(\ell = 1 \,|\, \mathbf{x}, \mathbf{y}_0'', \mathbf{y}_1') \right)} \right] d\mathbf{x}. \tag{45}$$

Consequently, assuming the BT assumption holds in the preference environment, the expected win rate improvement for the target of RLHF/DPO is as follows:

$$\text{Win Rate}_{p(\mathbf{y}_0 \,|\, \mathbf{x})}[p_{\text{RLHF/DPO}}(\mathbf{y} \,|\, \mathbf{x})]$$
$$= \int p(\mathbf{x}) \left[ \frac{\mathbb{E}_{p(\mathbf{y}_1 \,|\, \mathbf{x})} \left[ \mathbb{E}_{p(\mathbf{y}_0 \,|\, \mathbf{x})} p(\ell = 1 \,|\, \mathbf{x}, \mathbf{y}_0, \mathbf{y}_1) \exp\left( \frac{1}{\beta} \mathbb{E}_{p(\mathbf{y}_0' \,|\, \mathbf{x})}[\text{logit } p(\ell = 1 \,|\, \mathbf{x}, \mathbf{y}_0', \mathbf{y}_1)] \right) \right]}{\mathbb{E}_{p(\mathbf{y}_1' \,|\, \mathbf{x})} \exp\left( \frac{1}{\beta} \mathbb{E}_{p(\mathbf{y}_0'' \,|\, \mathbf{x})}[\text{logit } p(\ell = 1 \,|\, \mathbf{x}, \mathbf{y}_0'', \mathbf{y}_1')] \right)} \right] d\mathbf{x}. \tag{46}$$

For completion, we write the expected win rate improvement for SFT in the same form, connecting the expressions in this section to the result of Theorem 5.1:

$$\text{Win Rate}_{p(\mathbf{y}_0 \,|\, \mathbf{x})}[p_{\text{SFT}}(\mathbf{y} \,|\, \mathbf{x})]$$
$$= \int p(\mathbf{x}) \left[ \frac{\mathbb{E}_{p(\mathbf{y}_1 \,|\, \mathbf{x})} \left[ \mathbb{E}_{p(\mathbf{y}_0 \,|\, \mathbf{x})} p(\ell = 1 \,|\, \mathbf{x}, \mathbf{y}_0, \mathbf{y}_1) \mathbb{E}_{p(\mathbf{y}_0' \,|\, \mathbf{x})}[p(\ell = 1 \,|\, \mathbf{x}, \mathbf{y}_0', \mathbf{y}_1)] \right]}{\mathbb{E}_{p(\mathbf{y}_1' \,|\, \mathbf{x})} \mathbb{E}_{p(\mathbf{y}_0'' \,|\, \mathbf{x})}[p(\ell = 1 \,|\, \mathbf{x}, \mathbf{y}_0'', \mathbf{y}_1')]} \right] d\mathbf{x}. \tag{47}$$

## J. Comparing methods

In Table 3, we compare various preference learning methods along the axes of target distribution, objective, and the method of estimating the preference classifier. Ideally, we want a method's target solution is able to put more probability mass over the more preferred sequences; its objective directly seeks to approximate that target over some other goal; and its estimate of the preference classifier is as accurate as possible.

Table 3: Comparison of preference learning algorithms along three dimensions of design choices: target distribution, objective, and estimation of the preference classifier. For readability, we substitute the preference classifier distribution $p(\ell = 1 | \mathbf{x}, \mathbf{y}_0, \mathbf{y})$ with $p_{\text{clf}}$.

|  | Target (unnormalized) | Objective (per-query) | Preference Classifier |
| --- | --- | --- | --- |
| WRO-KL | $p(\mathbf{y} \,|\, \mathbf{x}) \exp(\frac{1}{\beta} \mathbb{E}_{p(\mathbf{y}_0|\mathbf{x})}[h \cdot p_{\text{clf}}])$ | $\text{KL}(p_\theta(\mathbf{y} \,|\, \mathbf{x}) \,\|\, p^*(\mathbf{y} \,|\, \mathbf{x}))$ | $\hat{p}_{\text{clf}}$ |
| RLHF | $p(\mathbf{y} \,|\, \mathbf{x}) \exp(\frac{1}{\beta} \mathbb{E}_{p(\mathbf{y}_0|\mathbf{x})}[\text{logit } p_{\text{clf}}])$ | $\text{KL}(p_\theta(\mathbf{y} \,|\, \mathbf{x}) \,\|\, p^*(\mathbf{y} \,|\, \mathbf{x}))$ | $\hat{r}(\mathbf{x}, \mathbf{y})$, BT |
| DPO | $p(\mathbf{y} \,|\, \mathbf{x}) \exp(\frac{1}{\beta} \mathbb{E}_{p(\mathbf{y}_0|\mathbf{x})}[\text{logit } p_{\text{clf}}])$ | $\text{KL}(p^*_{\text{clf}} \,\|\, p_{\theta,\text{clf}})$ | Equation (4), BT |
| SFT | $p(\mathbf{y} \,|\, \mathbf{x}) \mathbb{E}_{p(\mathbf{y}_0|\mathbf{x})}[p_{\text{clf}}]$ | $\text{KL}(p^*(\mathbf{y} \,|\, \mathbf{x}) \,\|\, p_\theta(\mathbf{y} \,|\, \mathbf{x}))$ | Bypass |

# K. Judge model

To train the judge model, we finetune a Pythia-2.8b base model using the same pairwise training set used to train all SFT models by simply modifying the prompt to take in $\mathbf{x}, \mathbf{y}_0, \mathbf{y}_1$ into account. The prompt template is as follows:

```
\n\n Human:  + <Instruction> +
\n\n Candidate Response A: + <response a> +
\n\n Candidate Response B: + <response b> +
\n\n The better answer is Candidate Response
```

The model is trained with a language modeling loss on the output, which is either 'A' or 'B'. For each pair of responses $\mathbf{y}_0, \mathbf{y}_1$, the training set includes two rows, one where $\mathbf{y}_0$ is Response A, and one where $\mathbf{y}_1$ is Response A.

The model is trained using RMSProp with a learning rate of 5e-7 and batch size of 64. The model is trained with a maximum sequence length of 512 and a maximum input length of 511. On the evaluation dataset, the model achieves a per-row classification accuracy of 68.8. (Training the same judge model with a sequence length of 1024 achieves the same accuracy, so we choose to stick with 512 for efficiency.)

To obtain the preference probability of a pair of outputs, we run a forward pass through the judge model twice, once with each order of output pairs, and average the results.

To simulate more opinionated preferences in this preference environment, we sharpen the judge preference probabilities with temperature scaling ($T = 0.2$) on the logit-transformed probabilities: $p(\ell = 1 \,|\, \mathbf{x}, \mathbf{y}_0, \mathbf{y}_1) = \sigma\left(\text{logit}\,\hat{p}_{\text{judge}}(\ell = 1 \,|\, \mathbf{x}, \mathbf{y}_0, \mathbf{y}_1)/T\right).$

# L. Experiment details

**Dataset processing.** For the Open Assistant dataset (Kopf et al., 2023), we keep only the first turn in each conversation and English-only examples, following Yuan et al. (2024). The dataset only has a train and validation split, so we split the original train set into a train and validation set and leave the validation set for testing / evaluation. The dataset includes multiple candidate responses for each input, all ranked, and to match the pairwise preference learning setup, we create a dataset of all possible pairs for each input. For instance, for a given input with three candidate responses (A, B, C), our paired dataset includes all three pairs (AB, BC, AC). For SHP, we keep only pairs where the score ratio is $> 2$, following Rafailov et al. (2024c).

**Training the initial model.** For each dataset, we finetune the base Pythia-2.8b models on all outputs, preferred and dispreferred. The resulting finetuned models serve as our initial models for preference learning. To train these models, we utilize a batch size of 64 and learning rate of 5e-7 chosen based on hyperparameter sweep between [1e-8, 5e-8, 1e-7, 5e-7, 1e-6] on OASST. Following Rafailov et al. (2024c), we use the RMSProp optimizer with a learning rate warm up of 150 steps and constant learning rate schedule otherwise. We evaluate every 100 steps and choose the best checkpoint based on validation loss.

**SFT and DPO experiments.** SFT and DPO experiments follow the same training configuration as the initial model.

**RL experiments.** We use the implementation of reward model training and PPO from the TRL library (von Werra et al., 2020). For reward model training, we use a batch size of 64, learning rate of 5e-7 for Pythia2.8b, and checkpoint every 100 steps, matching the SFT and DPO experiments. For PPO, we use a learning rate=1e-6 (obtained through a hyperparameter sweep of [1e-7, 5e-7, 1e-6] on OASST), batch size=128, and PPOConfig defaults for all other hyperparameters. We checkpoint every five steps and choose checkpoint with the best policy loss (namely, ignoring the estimation of the value head).

**Win rate evaluations.** We sample a set of 100 input prompts from the test set of a given dataset (same 100 prompts for all models) and perform win rate evaluation using the oracle judge for the dataset.

## M. Exploring optimization benefits of RLHF

Below, we compare the win rate results when optimizing the RLHF objective versus a non-optimized WRO-KL-BT-logit objective that keeps the $\mathbb{E}_{p(\mathbf{y}_0 \mid \mathbf{x})} r(\mathbf{x}, \mathbf{y}_0)$ term. Namely, the RLHF objective we optimize is

$$-\mathcal{L}_{\text{RLHF}}(\theta) = \max_{\theta} \mathbb{E}_{p(\mathbf{x})} \left[ \mathbb{E}_{p_\theta(\mathbf{y} \mid \mathbf{x})}[r(\mathbf{x}, \mathbf{y})] - \beta \text{KL}(p_\theta(\mathbf{y} \mid \mathbf{x}) \,\|\, p_{\text{ref}}(\mathbf{y} \mid \mathbf{x})) \right]. \tag{48}$$

The non-optimized WRO-KL-BT-logit objective we optimize is

$$-\mathcal{L}_{\text{RLHF}}(\theta) = \max_{\theta} \mathbb{E}_{p(\mathbf{x})} \left[ \mathbb{E}_{p_\theta(\mathbf{y} \mid \mathbf{x})}[r(\mathbf{x}, \mathbf{y}) - \mathbb{E}_{p(\mathbf{y}_0 \mid \mathbf{x})} r(\mathbf{x}, \mathbf{y}_0)] - \beta \text{KL}(p_\theta(\mathbf{y} \mid \mathbf{x}) \,\|\, p_{\text{ref}}(\mathbf{y} \mid \mathbf{x})) \right]. \tag{49}$$

Win rate results can be found in Table 4. All experimental details match that of the main paper.

Table 4: Comparison of win rates between RLHF (optimized) and WRO-KL-BT-logit without additional optimization (non-optimized). RLHF's dropping of the constant in the objective decreases variance in the gradient estimates and generally improves results with small enough $\beta$. However, this change does not yield systematic benefits for every setting, suggesting that there is still room for improvement.

| Dataset | | $\beta = 0.001$ | $\beta = 0.01$ | $\beta = 0.1$ | $\beta = 1$ |
|---|---|---|---|---|---|
| HH | non-optimized | 51.79 (1.12) | 54.59 (0.90) | 68.68 (0.80) | **56.88 (0.46)** |
| | optimized | **70.46 (0.87)** | **63.93 (1.24)** | 69.44 (0.90) | 54.60 (0.67) |
| OASST | non-optimized | 60.74 (3.65) | 61.79 (3.64) | **73.11 (1.16)** | 60.41 (2.05) |
| | optimized | 62.05 (3.65) | 63.95 (3.38) | 66.72 (1.16) | 58.45 (1.82) |

