# OpenReview forum: "Preference learning made easy: Everything should be understood through win rate"
_ICML.cc/2025/Conference — ICML 2025 poster_

### Official Review · Reviewer_F6Z7 · 2025-02-17

**Overall Recommendation:** 3

**Summary:**

The paper first introduce the concepts of preference consistency and prevalence consistency, and then proved that the only form of loss function that satisfies both preference and prevalence consistency is a type of win rate, proposing the h-win rate. The paper then argues the benefits of h-win rate, and analyzed DPO and SFT through this lens, showing that they do not fit the consistencies proposed. Empirical results shows a correspondance between win rate and the loss proposed.

**Claims And Evidence:**

yes. proofs look correct and empirical results indicate relavance.

**Essential References Not Discussed:**

no.

**Experimental Designs Or Analyses:**

The experiments, while well indended, look a bit small in scale and not very strong. In particular, there are three sets of experiments. Figure 2 compares expected and actual win rates of different methods, but it's not clear the datasets and the model (I assume it's pythia 2.8B and Anthropic HH, but unclear); Table 1 shows different h win rate methods, but did not show a clear winner, and thus it's hard to draw a useful conclusion there; Figure 3 shows the correspondance between train loss and win rate, but I'm curious about the sudden drop of loss on the left subfigure.
In general, while the idea is interesting and useful, the experiments are not well supported, it would be helpful to conduct higher scale experiments, and show better correspondance between win rates and train loss, which is the main benefit of the methodology part.

**Methods And Evaluation Criteria:**

The paper mostly proposes two criteria to look at when analyzing alignment algorithms. Well in experiments the resulting algorithm from the theory does not seem to outperform existing methods, it does help elucidate why DPO often do not achieve SOTA results in practice.

**Other Comments Or Suggestions:**

no

**Other Strengths And Weaknesses:**

no

**Questions For Authors:**

Is it possible to extend the proposed pairwise metrics to listwise? Practitioners often have a list of N_i responses to the i-th prompt, and preference scores assigned to each of the responses.

**Relation To Broader Scientific Literature:**

The paper helps elucidate some of the alignment research problems in practice.

**Theoretical Claims:**

mostly checked prop 3.3 and skimmed the others. Looks correct.

---

> ### Author Rebuttal · Authors · 2025-04-01
>
> Thank you for your review. We answer your questions (paraphrased for brevity) below:
>
> 1. [datasets and model of Figure 2?]
> -  The reviewer is correct that the model is Pythia 2.8B. The dataset is Open Assistant.
> 2. [Table 1 shows different h win rate methods, but did not show a clear winner, and thus it's hard to draw a useful conclusion there]
> - We'd like to emphasize that the lack of a clear winner is the primary conclusion of Table 1. This result suggests that  design choices for WRO which affect the optimal solution do not seem to have a first-order systematic effect on performance in practice;  optimization success does though, as a show later in Figure 3.
> 3. [Figure 3 shows the correspondence between train loss and win rate, but I'm curious about the sudden drop of loss on the left subfigure.]
> - Great question. Each point on either plot in figure 3 is a different final model from a training run of a given WRO variant. The sudden drop of loss denotes a subset of models which achieve final loss past a certain threshold (around -0.35). These models achieve much better win rates than models which are not able to reach such a loss value, suggesting that simply being able to optimize past a certain threshold is strongly indicative of test win rate performance.
> 4. [The experiments, while well intended, look a bit small in scale and not very strong. It would be helpful to conduct higher scale experiments, and show better correspondence between win rates and train loss, which is the main benefit of the methodology part.]
> - We'd like to emphasize that the current experiments consist of 72 distinct RL fine-tuning runs on a several billion parameter model. Unfortunately, limited resources prevent us from significantly scaling up past the current experiments during the response period, but if the reviewer has specific suggestions, we'd be happy to incorporate them into the final version of the paper.
> - It is also worth noting that the level of correspondence we are already seeing between train loss and test win rate is extremely striking, especially given the fact that different WRO variants do not necessarily even share the same scale for loss (due to choice of $\beta$ and $h$). In other words, we would not expect, a priori, such significant rank correlations, much less a better correspondence. Instead, the fact that the correlation exists with train loss, even when it doesn't with other design choices we hypothesized a priority to be relevant, suggests that optimization success is the first order consideration for WRO.
> 5. [Is it possible to extend the proposed pairwise metrics to listwise?]
> - Yes! In the case of preference for a set of $k$ responses, the binary preference classifier would become a k-class preference classifier, and h-win rate would become a multi-way win rate over $k-1$ anchor distributions, which could optionally be the same distribution. Additional assumptions also generalize, such as Plackett-Luce for Bradley-Terry. We’ve added this discussion to the appendix.
>
> Thank you for your review. Please let us know if you have any additional questions or concerns; otherwise, we would greatly appreciate it if you would consider raising your score.

---

### Official Review · Reviewer_7gys · 2025-03-14

**Overall Recommendation:** 3

**Summary:**

The paper examines what constitutes a "grounded" evaluation of a policy or language model's alignment with (human) preferences. Assuming the evaluation function is both preference-consistent and prevalence-consistent, meaning it is linear in the distribution of contexts, the distribution of alternatives, and the policy under evaluation, the paper establishes that the only grounded evaluation metric is the $h$-win rate.

Building on this, the paper argues that maximizing the $h$-win rate (for some $h$) is a natural objective for aligning generative models. This approach offers two key benefits: (1) win rate correspondence, meaning that optimizing the objective directly improves the $h$-win rate, and (2) win rate consistency, ensuring that the optimal policy obtained from the optimization also maximizes the $h$-win rate. RLHF is one such win rate optimization (WRO) algorithm. However, methods like DPO and SFT do not optimize win rates. In particular, DPO lacks win rate correspondence.

The paper then presents an empirical study to test whether theoretical insights translate to practice and whether WRO methods outperform others. Interestingly, the empirical results contradict theoretical expectations, highlighting the importance of optimization and other influencing factors.

**Claims And Evidence:**

The paper's main claims are theoretical. Given the proposed properties of the evaluation function, namely, preference-consistency and prevalencepconsistency, are desirable, the paper then provides a sound theoretical foundation. Additionally, the authors are transparent about the empirical evaluations, which I appreciate.

There is a subtlety in interpreting the results that I wish the authors had discussed further. Specifically, the dependence of the optimal solution on the anchor distribution is, by design, undesirable. This issue affects all $h$-win rate methods except those using BT with $h = \text{logit}$ (i.e., RLHF). It is counterintuitive that what makes a generative model optimal depends on the anchor distribution used for comparison. Compare this to reward maximization where the optimal policy does not depend on how the preference dataset is sampled. For further discussion, see [2] in the additional related work.

**Essential References Not Discussed:**

The paper has done a good job of covering relevant literature. When I heard the win rate, I immediately thought of two very recent works from social choice theory and alignment [1, 2]. These two also discuss how RLHF and DPO are related to win rate, or in other words, Borda count. Their setting is slightly different from the current work as they consider the possibility of different reward functions for individuals, but I thought they can still be interesting and relevant for the authors.
[1] Distributional preference learning: Understanding and accounting for hidden context in RLH
[2] Direct Alignment with Heterogeneous Preferences

**Experimental Designs Or Analyses:**

I appreciate the transparency of the results. I think the experiments were insightful and interesting. A few minor questions: Please further elaborate on how three cases for $\hat{p}_l$ are obtained in Table 1. How did you conclude from experiments that SFT > DPO? My read of Fig. 2 is that DPO is superior to all methods.

Another room for improvement is Fig. 1. I suspect some of the win rate correspondence violations of DPO are due to its KL-regularization. I think Fig. 6 partly confirms this, though this cannot completely explain why DPO is not win rate correspondent. Compared to Fig. 6, I think Fig. 1 is a very extreme case. I'd choose a smaller $\beta$. Also, it helps to have the same plot for logit-WRO-KL so we can see how many of these violations are due to regularizations.

**Methods And Evaluation Criteria:**

Again, assuming that the evaluation function should be preference-consistent and prevalence-consistent, WRO makes sense. However, I wish there were better motivations for these properties. In particular, as I understand, prevalence-consistency for the generator basically rules out non-RL methods, including all direct methods (that's my read from Eq. 2 where I don't have any other way than RL to maximize w.r.t. $\theta$).

**Other Comments Or Suggestions:**

N/A

**Other Strengths And Weaknesses:**

The paper is well-written. Just as a suggestion, maybe use consistent capitalization in the title and sections.

**Questions For Authors:**

Please refer to above points.

**Relation To Broader Scientific Literature:**

The paper mostly reminds me of $\psi$-PO (Azar et al.) but still has new insights. It is also very related to GPO (Tang et al.). All of these works try to understand preference optimization in its most generality. I like how this work further categorizes these methods in terms of WRO and non-WRO.

**Theoretical Claims:**

As far as I checked, the theory is sound. I think you don't need BT to prove Prop. 4.1. I suspect any increasing symmetric $sigma$ works here. Another suggestion is that having two anchor/competitor distributions was confusing. Since you will assume these are the same, maybe do not introduce them differently.

The analysis of DPO was interesting. First, I think DPO is win rate-consistent, which could be better highlighted ([1] and [2] have similar observations, I believe). Second, this analysis made me think if win rate-correspondence is a big deal. Doing a little exploration, I came across this intuition: calculating the gradient of the DPO's objective, we can see that DPO works to make the win rate induced by a policy get closer to the true win rate under true reward. When these two completely match, DPO's policy internally represents the true reward, which means it is the optimal policy. However, this does not mean that we are getting closer to the optimal reward in every step. This is only my intuition but it would help if the authors clarify/elaborate further on their intuition in Sec 5.1. I think this will be very helpful as the readers generally don't expect and MLE estimator such as DPO to have an undesirable property.

---

> ### Author Rebuttal · Authors · 2025-04-01
>
> Thank you for the thoughtful review! Questions (paraphrased) and responses below:
> 1. [Undesirable that optimal solution depends on anchor, not true for reward maximization]
> - This dependence is a limitation of the information contained in the distribution of pairwise comparisons itself. As mentioned in Prop 4.1, we agree that the issue can be bypassed with additional assumptions (e.g., under BT and finite rewards, optimizing any ℎ-win rate over any anchor optimizes for all ℎ-win rates over all anchors). We have added more discussion.
> 2. [Better motivations for properties, prevalence-consistency for generator rules out non-RL]
> - We want preference-consistency or else a model can be deemed good even if it generates dispreferred responses. We want prevalence-consistency wrt generator or else we are not evaluating the model’s generation behavior. Enforcing prevalence-consistency means objectives include an expectation over the model, but this could be approximated with off-policy samples and importance weighting for instance.
> 3. [I don’t think you need BT to prove Prop 4.1]
> - Good point. Prop 4.1 is meant to show what BT gives us (see 1), but we have generalized the proof.
> 4. [Two anchor/competitor distributions is confusing]
> - Could you clarify what you mean? If you mean $p_{anchor}$ vs. $p_{ref}$, we distinguish them to highlight that the anchor we are optimizing against can be different from the reference we are regularizing to (but please let us know if we are misunderstanding!)
> 5. [Highlight that DPO is win rate-consistent]
> - DPO’s optimal solution matches that of a regularized WRO objective; thus we can say that it satisfies regularized win rate consistency. We mention this in appendix A and have added a line in the main paper, but it is also the motivation for DPO, i.e., an objective whose solution matches that of RLHF.
> 6. [Is win-rate correspondence a big deal? Can you clarify the intuition for DPO?]
> - Failing win rate correspondence makes training and model selection difficult (i.e., why reduce loss if it doesn’t help with the goal, how to select checkpoint when best loss is not best win rate).
> - Intuition: DPO performs MLE of the preference classifier, where this classifier is a function of the policy model. If loss goes down, this implicit preference classifier is getting closer to the true preference classifier (in expectation over the offline distribution). However, this does not mean that the corresponding policy model is getting better at win rate under the true preference classifier. First, prevalence mismatch between offline data and model means the preference classifier could get worse for responses that are relevant under the model even as it gets closer to ground truth for the offline data. Second, DPO breaks preference consistency as the implicit preference classifier is a function of the policy model itself and thus changes throughout training as the policy model changes.
> 7. [Elaborate on three cases for $p_{\ell}$]
> - Oracle is a model used to label the preference data and evaluate the win rate of the trained models. BT = True is a reward model trained on the oracle-labeled preference data. BT = False is a preference classifier trained on the oracle judge-labeled preference data without BT (lines 377-384).
> 8. [SFT > DPO?]
> - Summarizing footer 4, we conclude SFT > DPO with respect to ease of optimization but based on works which show optimization difficulties of DPO, e.g., failure to increase the probability of the chosen response or improve the rankings.
> 8. [I suspect some win rate correspondence violations of DPO are due to KL, Fig 1 is extreme]
> - To formalize, we add a new property called regularized win rate correspondence, i.e., improvement in loss implies improvement in win rate or divergence to reference. This is very inclusive, as even a change that results in a large decrease in win rate will still meet regularized win rate correspondence if divergence to reference decreases. Any regularized WRO objective satisfies this. DPO (off- or online) does not: loss can improve even though win rate and divergence get worse (as DPO does not directly optimize for either).
> - Note that Fig 1 is less extreme than DPO as setting is online and infinite data. Fig 5 is even less extreme with a uniform starting model and equally spaced rewards, and Fig 6 further decreases $\beta$ in the least extreme setting in Fig 5. These figures show that DPO still does not satisfy win rate correspondence even as we continue to make the setting more and more favorable (and at some point, smaller $\beta$ doesn't help). Even so, given that the Fig 1 is quite specific, we've moved it to the appendix and have replaced it with a discussion on regularized win rate correspondence.
> 10. [Additional related work]
> - Thank you, they were a pleasure to read, and we have added them to related work.
>
> Thank you again for your questions. If we have answered them to your satisfaction, we hope you will consider raising your score.

---

### Official Review · Reviewer_1f3H · 2025-03-14

**Overall Recommendation:** 3

**Summary:**

This paper introduces two consistency measures to study preference models. The paper proves that the only evaluation criteria that respects both is the win-rate. This finding is generalized into h-WinRate -- win rate under a monotonically non-decreasing transformation $h$. This measure is used as an optimization objective, called win rate optimization (WRO), which is later used to study preference learning algorithms. The paper shows that while RLHF respects this objective (under KL-regularization), DPO and SFT do not. Empirical analysis of DPO exhibits that improving DPO loss does not necessarily improve the win-rate. Further experiments with varying $h$, $\beta$ and preference model suggests there is no one winning setting across different benchmarks.

**Claims And Evidence:**

The paper claims that win-rate is the unique measure that should be used to evaluate preference models. This is also supported by other work in literature, including IPO or BonBon. The paper hypothesizes two consistency measures that uniquely suggest win-rate is the only evaluation that respects these measures. These claims are supported by theoretical analysis of RLHF, DPO, and SFT.

The paper empirically shows that DPO-loss is not necessarily correlated with win-rate, which is suggested by the theory. But the remaining experimental results are mixed.

- One main criticism is that the theory in the paper is similar to IPO, especially the optimization and analysis of preference learning. Similar to the paper, IPO uses a non-decreasing function with win-rate. The proposition 4.1 can be proven using Eq (7) in IPO with $\tau$ going to zero. I think having more empirical analysis can distinguish the paper better than related work.

- Expected win-rates suggest a global ordering amongst RLHF, DPO, and SFT, with a monotonically non-decreasing relationship between win-rate and $\beta$. But, empirical results suggest otherwise; DPO performs the best and RLHF is slightly better than SFT. While the authors suggest there might be other factors at play, such as optimization etc. these are not supported by any empirical evidence.

- It is interesting to me that using oracle $\hat{p}_l$ doesn't exhibit the best performance. Given that the true objective is to improve the win-rate under the oracle preferences, this is counter-intuitive.

- The paper uses only Pythia-2.8b model to train preference models as well as to judge results. I think this limits the analysis. Using the same base model for both preference learning and judging can introduce biases. It would help if other similar sized models, such as Gemma2-2b, Qwen2-1.5b, are used as well.

**Essential References Not Discussed:**

Related work sufficiently covers the literature.

**Experimental Designs Or Analyses:**

As I explained above, experimental results are limited. I think using other similar sized LLMs, analyzing the results in more detail would help the paper.

**Methods And Evaluation Criteria:**

The paper studies win-rate through newly-proposed measures and introduces a new objective for preference learning. It compares different methods under win-rate; which is the main metric of study. There are two datasets that are commonly studied in the literature.

**Other Comments Or Suggestions:**

One minor comment is about the notation. Functional composition in section 4.2 is different from the rest. Please use a consistent notation.

**Other Strengths And Weaknesses:**

I think proposition 3.3 and definition 3.2 can be practically useful to check whether a preference learning method respects the win-rate objective. The alternative, comparing the objective directly to win-rate objective, can be more challenging.

**Questions For Authors:**

Please see above respective sections for related questions.

**Relation To Broader Scientific Literature:**

The paper is broadly relevant to preference learning.

**Theoretical Claims:**

I checked the correctness of proofs in the main text.

---

> ### Author Rebuttal · Authors · 2025-04-01
>
> Thank you very much for your review. We answer your questions (paraphrased for brevity) below:
>
> 1. [Theory in the paper is similar to IPO. Proposition 4.1 can be proven using equation 7 in IPO with $\tau$ going to zero. More empirical analysis can distinguish the paper.]
> - We agree that one can use equation 7 in the IPO paper to gain intuition for proposition 4.1, but equation 7 in the IPO paper is the target distribution specifically for KL-regularized WRO objectives and does not include the additional analysis required to take the limit of $\tau$ to zero (e.g., proving existence of a limit, connection between limit of solution and solution of limit`)  nor the specific implication of the Bradley Terry assumption. In contrast, Proposition 4.1 in this work  is meant to emphasize a consequence of the Bradley Terry assumption on the dependence of WRO on choice of anchor distribution: Namely, it proves that under the Bradley-Terry assumption and finite rewards, the resulting solution optimizes win rate over all possible anchors regardless of the choice of anchor in the objective. In other words, the Bradley-Terry assumption effectively gives us the ability to optimize over all anchors, whereas without that assumption we choose the anchor. We have updated the text to emphasize this point.
> 2. [Expected versus observed win rates; the authors suggest there might be other factors at play, such as optimization etc., but these are not supported by any empirical evidence.]
> - Section 6.3 / Figure 3 provides empirical evidence of optimization being an important factor for the misalignment between expected versus observed win rate patterns. Namely, the train loss achieved by a given run is significantly predictive of the test win rate, even as other factors which dictate the target distribution of the objective are not (Table 1). This result is especially striking given that the losses for different objectives do not necessarily share the same scale.
> 3. [It is interesting to me that using oracle $\hat{p}_\ell$  doesn't exhibit the best performance]
> - We agree. This result suggests that there is a more important first order factor, which Figure 3  (a scatterplot of train loss versus test win rate for selected models across different WRO runs), and the corresponding significance test suggests is optimization success.
> 4. [The paper uses only Pythia-2.8b model to train preference models as well as to judge results. I think this limits the analysis. Using the same base model for both preference learning and judging can introduce biases. It would help if other similar sized models, such as Gemma2-2b, Qwen2-1.5b, are used as well.]
> - Due to resource constraints, we are unfortunately unable to run the same experiments on additional models at this time (requires 7 offline finetuning jobs and 36 online finetuning jobs for just a single dataset with no hyperparameter sweep), but we take the reviewer's point that it would be helpful to add more models to the final paper and are working to do so. And just to clarify, we are using separately trained models for the judge and policy model even if they are based on the same base model.
> 5. [I think proposition 3.3 and definition 3.2 can be practically useful to check whether a preference learning method respects the win-rate objective. The alternative, comparing the objective directly to win-rate objective, can be more challenging.]
> - Thank you for highlighting the practical usefulness of proposition 3.3 and definition 3.2. We agree!
> 6. [Section 4.2 notation]
> - Thanks for the feedback, fixed!
>
> We hope we have addressed your concerns, and we hope you will consider raising your score. Thank you!

---

> > ### Comment · Reviewer_1f3H · 2025-04-05
> >
> > Thank you for the clarification.
> >
> > 1. Regarding Eq 7 in IPO paper, it is under the assumption of a BT model: "Applying this proposition to the objective function of
> > Equation (6), for which there exists an analytical solution, reveals that under the BT assumption the closed form solution to DPO and RLHF can be written as". For $\tau$, I agree with you, it seems the authors in IPO assumed it is strictly positive for their analysis.
> >
> > 2. For Figure-3 to support your claim, worse training loss should indicate almost an ordering amongst different methods, like: SFT > RLHF > DPO. Can you add colors to Figure-3 to understand model-based correlations? Can you explain if these losses are comparable across models?

---

> > > ### Author Response · Authors · 2025-04-05
> > >
> > > Thanks for the comments!
> > > 1. We agree that the IPO paper the mentions the BT assumption to connect Eq 7 with RLHF & DPO. The additional implication of the Bradley Terry assumption highlighted in Proposition 4.1 in our work is that under the assumption and finite rewards, all WRO methods share the same optimal solution which is also optimal over all anchor distributions.
> > > 2. Thanks for the question. To clarify, figure 3 only plots WRO runs (which include RLHF), not any SFT or DPO runs. The figure is meant to showcase that optimization success is a first-order of consideration for WRO. We would not expect SFT & DPO losses to be directly comparable to each other or RLHF (for instance, neither SFT nor DPO satisfy win rate correspondence); in fact, we would not necessarily expect WRO losses across different values of h or beta to be comparable either, making the fact that there is a trend between loss and win rate across these runs all the more notable. See [here](https://imgur.com/a/Ntu6Her) for Figure 3 stratified into the different choices of $h$, $\beta$, $\hat{p}_\ell$.

---

### Official Review · Reviewer_nt27 · 2025-03-17

**Overall Recommendation:** 2

**Summary:**

The paper argues that win rate should be the primary evaluation metric in preference learning, as it is the only measure that respects both preferences and prevalences in pairwise comparison data. The authors introduce a win rate-centric framework and classify preference learning methods into WRO and non-WRO approaches.

**Claims And Evidence:**

Yes.

**Essential References Not Discussed:**

None.

**Experimental Designs Or Analyses:**

Yes,

**Methods And Evaluation Criteria:**

Yes.

**Other Comments Or Suggestions:**

None.

**Other Strengths And Weaknesses:**

None.

**Questions For Authors:**

None.

**Relation To Broader Scientific Literature:**

The authors’ analysis is well-reasoned; however, I believe their theoretical framework relies on overly idealized assumptions, which diminishes the practical significance of their conclusions. I agree that win rate is the most aligned metric with preference, but the analysis overlooks key challenges:

1. In-Distribution Assumption: The authors assume that training and evaluation data are drawn from the same distribution. In practice, training data cannot fully cover real-world applications, necessitating regularization to improve OOD generalization.

2. Unreliable Preference Models: The preference model may be biased or unreliable, which is why additional optimization objectives—such as length regularization—are incorporated into preference optimization. While these objectives break WRO, they often lead to better empirical results.

Given these limitations, I disagree with the claim that "everything should be understood through win rate.", and WRO methods, as acknowledged by the authors, not necessarily outperform non-WRO ones. While the there are some theoretical insights, the conclusions provide limited practical guidance for real-world preference learning and this area.

**Theoretical Claims:**

Yes.

---

> ### Author Rebuttal · Authors · 2025-04-01
>
> Thanks for the review. Responding to your concerns below:
>
> 1. [Overly idealized assumptions (in-distribution assumption, unreliable preference models)  diminish practical significance of conclusion]
>   - Our framework is not focused on idealized assumptions per se but rather what can be learned from the preference data distribution alone, specifically to clarify the role of assumptions vs. information inherent in preference data itself. In fact, our analysis is meant to make it easier to consider challenges such as those posed by the reviewer: Namely, our work clarifies what is optimal when nothing else is considered besides the information contained in preference data alone. This understanding of what is in preference data makes it possible to disentangle anything extra in a given method as an additional assumption or strategy. We are not arguing that anything extra is bad if the resulting objective is not pure WRO; instead, win rate optimization should be seen as the starting point, with any addition or modification understood modularly based on the underlying assumption being encoded.
>   - To drive that point home, we've added the following lines to the discussion:
>     - …this work offers a simplifying insight: win rate is the only evaluation that can matter based on preference data alone, and thus all of preference learning should be understood in relation to it—both how well a given method optimizes for win rate as well as the role of additional assumptions that move a method beyond pure WRO.
>
> 2. [Limited practical guidance for real world preference learning]
>   - We respectfully disagree; some immediately implementable practical guidance that comes out of our analysis includes 1. given how important and finicky optimization success is for WRO objectives (including RLHF), it could be helpful to kick off multiple seeds to find one that optimizes best; 2. given that DPO loss fails win rate correspondence on multiple fronts, one should consider performing model selection with a metric other than validation loss; 3. greater generation diversity as well as alternative filtering strategies can improve the win rate limits of supervised fine tuning on preferred samples; 4. studying optimization strategies (e.g.,  contemporaneous paper [1]) would be a high leverage direction of inquiry, given the importance of optimization success in practical performance of preference learning algorithms.
>
> Based on our response, we hope you will consider raising your score. Thank you!
>
> [1] https://arxiv.org/pdf/2503.14286

---

### Decision · Program_Chairs · 2025-05-01

**Decision:**

Accept (poster)

**Comment:**

This paper presents a well-motivated and theoretically grounded framework for understanding preference learning through the lens of win rate optimization (WRO). By formalizing the concepts of preference and prevalence consistency, the authors demonstrate that win rate is the only evaluation metric that faithfully reflects the underlying preference data distribution. The work introduces a useful taxonomy distinguishing WRO and non-WRO methods, and provides both theoretical and empirical analyses to support its claims. While one reviewer expressed concerns about the idealized assumptions and practical implications, the authors’ rebuttal offered thoughtful clarifications and highlighted concrete guidance for real-world applications. Overall, this paper offers valuable conceptual clarity to the preference learning literature and advances the foundation for future research in alignment. I recommend acceptance.